# The caudate nucleus contributes causally to decisions that balance reward and uncertain visual information

Takahiro Doi[1,2], Yunshu Fan[1,3], Joshua I Gold[1,3], Long Ding[1,3]*

[1]Department of Neuroscience, University of Pennsylvania, Philadelphia, United States; [2]Department of Psychology, University of Pennsylvania, Philadelphia, United States; [3]Neuroscience Graduate Group, University of Pennsylvania, Philadelphia, United States

**Abstract** Our decisions often balance what we observe and what we desire. A prime candidate for implementing this complex balancing act is the basal ganglia pathway, but its roles have not yet been examined experimentally in detail. Here, we show that a major input station of the basal ganglia, the caudate nucleus, plays a causal role in integrating uncertain visual evidence and reward context to guide adaptive decision-making. In monkeys making saccadic decisions based on motion cues and asymmetric reward-choice associations, single caudate neurons encoded both sources of information. Electrical microstimulation at caudate sites during motion viewing affected the monkeys' decisions. These microstimulation effects included coordinated changes in multiple computational components of the decision process that mimicked the monkeys' similarly coordinated voluntary strategies for balancing visual and reward information. These results imply that the caudate nucleus plays causal roles in coordinating decision processes that balance external evidence and internal preferences.

**\*For correspondence:**
lding@pennmedicine.upenn.edu

## Introduction

Effective decision-making often requires deliberation over uncertain evidence for and against different alternatives, as well as over the expected outcome associated with those alternatives. The final choice depends on how these two types of information are combined in the deliberation process. For example, when we encounter news that "chocolate is healthy," our reaction ("should I eat more chocolate?") can be influenced by both the perceived quality of the evidence (peer-reviewed research article or tabloid) and our desired version of events ("I hope it is true because I love chocolate!"). Previous studies have provided many insights into the kinds of computations that humans and animals use to balance uncertain sensory evidence and reward expectation for adaptive decisions, but it remains unclear where and how these computations are implemented in the brain (*Diederich and Busemeyer, 2006*; *Fan et al., 2018*; *Feng et al., 2009*; *Gao et al., 2011*; *Leite, 2012*; *Liston and Stone, 2008*; *Maddox and Bohil, 1998*; *Mulder et al., 2012*; *Summerfield and Koechlin, 2010*; *Teichert and Ferrera, 2010*; *Voss et al., 2004*; *Waiblinger et al., 2019*; *Whiteley and Sahani, 2008*).

A prime candidate for mediating these computations is the basal ganglia pathway, which has been a focus of many modeling studies (*Bogacz and Gurney, 2007*; *Ding and Gold, 2013*; *Hikosaka et al., 2014*; *Kable and Glimcher, 2009*; *Lo and Wang, 2006*; *Rao, 2010*; *Ratcliff and Frank, 2012*; *Redgrave et al., 1999*; *Summerfield and Tsetsos, 2012*; *Wei et al., 2015*). This pathway is known to make separate contributions to perceptual decisions that select based on the interpretation of uncertain sensory evidence and value-based decisions that select among outcome options (*Amemori et al., 2018*; *Cai et al., 2011*; *Cavanagh et al., 2011*; *Ding and Gold, 2010*;

*Ding and Gold, 2012b*; *Hikosaka et al., 2014*; *Kim and Hikosaka, 2013*; *Kimchi and Laubach, 2009*; *Lau and Glimcher, 2008*; *Nakamura and Hikosaka, 2006b*; *Samejima and Doya, 2007*; *Santacruz et al., 2017*; *Seo et al., 2012*; *Tachibana and Hikosaka, 2012*; *Tai et al., 2012*; *Wang et al., 2018*; *Yanike and Ferrera, 2014*; *Yartsev et al., 2018*). However, its role in combining those different sources of information remains speculative. For example, bilateral lesions of the striatum in rats affected action vigor but not the decision process that combined reward and olfactory input (*Wang et al., 2013*), arguing against a casual role of the basal ganglia pathway.

To begin to understand the basal ganglia's causal roles in combining evidence and outcome expectation in guiding adaptive decision-making, we targeted the caudate nucleus, an input station in the oculomotor basal ganglia pathway, in monkeys trained to report their perceived motion direction of a random-dot kinematogram in an asymmetric-reward task (*Figure 1A*; *Fan et al., 2018*). Across trials, we manipulated visual evidence by presenting motion stimuli with varying strengths in two directions. Across blocks of trials, we manipulated reward context by assigning a large reward for one direction and small reward for the other and alternating reward assignments between two consecutive blocks. As we demonstrated previously, monkeys solved this task by adaptively balancing the uncertain visual input and reward information (*Fan et al., 2018*).

Here, we report four lines of evidence supporting a causal involvement of the caudate nucleus in mediating the monkeys' adaptive strategy: (1) caudate activity exhibited combined representations of task-relevant visual and reward information, both at the population level and at the single-neuron level; (2) electrical microstimulation in the caudate nucleus during motion viewing affected how the visual and reward information was used to form the decision, often in a reward context-dependent manner; (3) the reward context-dependent microstimulation effects shared certain features with the monkeys' voluntary, adaptive adjustments in response to the asymmetric reward contexts; and (4) the magnitude of microstimulation effects on these features depended on caudate activity patterns at the stimulation sites. These results imply that the caudate nucleus plays key roles in coordinating the decision process that balances external evidence and internal preferences to guide adaptive behavior.

## Results

We trained two monkeys to report their decision by making a saccadic eye movement at a self-determined time. As we reported previously, the monkeys' performance depended on both the strength and direction of the visual-motion evidence and the reward asymmetry on each trial, with consistent biases toward choices associated with large reward (*Figure 1B and C*). More details about these two monkeys' performance on this task can be found in *Fan et al., 2018*.

### Caudate neurons encode both visual and reward information

We first tested if and how individual caudate neurons encoded task-relevant visual and reward information. We predicted that, if the caudate nucleus contributes to integrating these different sources of information, then the activity of individual neurons would be modulated by properties of both the visual motion evidence and the reward context. Alternatively, if the caudate nucleus processes reward and visual information separately, then the activity of individual caudate neurons would be modulated by either type of information alone but not together.

We found that many caudate neurons are sensitive to both visual and reward information used to perform the task. Specifically, we recorded single-unit activity from 142 caudate neurons in the two monkeys (*n* = 49 for monkey C, 93 for monkey F). These caudate neurons showed diverse patterns of task-dependent modulations that in many cases included a combination of visual and reward modulations. For example, the activity of the neuron depicted in *Figure 2A* showed three types of modulation: (1) more activity during the blocks when the contralateral choice was paired with small reward and the ipsilateral choice was paired with large reward (green > purple); (2) more activity for trials with stronger versus weaker motion evidence (dark shade > light shade; i.e., higher versus lower coherence levels, respectively), particularly for trials with contralateral choices; and (3) more activity for trials with contralateral versus ipsilateral choices, both during motion viewing and around saccade onset (Contra > Ipsi). This neuron's activity thus reflected a combination of reward context, motion strength, and eventual choice. The example neuron depicted in *Figure 2B* showed: (1) more activity on trials with higher coherence levels (dark shade > light shade); (2) a contralateral choice

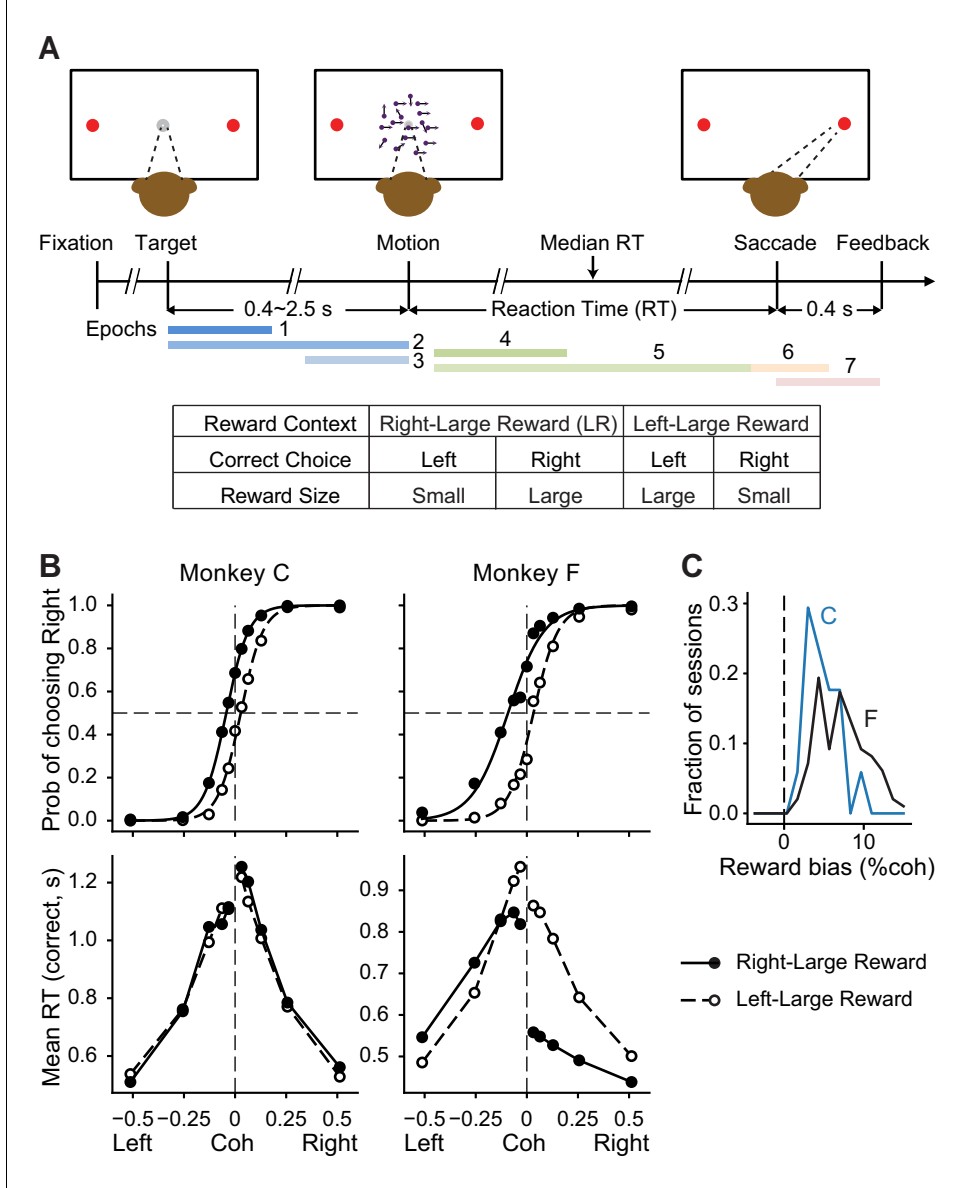

**Figure 1.** Monkeys biased toward choices associated with large reward. (**A**) Task design and timeline. Monkeys reported the perceived motion direction with saccades to one of the two choice targets. The motion stimulus was turned off upon detection of saccade. Correct trials were rewarded based on the reward context. Error trials were not rewarded. The color bars in the timeline indicate epoch definitions for the regression analysis of neural firing rates in *Equation 1*. (**B**) Average choice (top) and RT (bottom) behavior of two monkeys (n = 17,493 trials from 38 sessions for monkey C, 29,599 trials from 79 sessions for monkey F). Filled and open circles: data from the two reward contexts. Lines in top row: logistic fits. (**C**) Histograms of reward bias (half of the horizontal shift between the two reward contexts estimated from logistic fits) for the two monkeys. Positive values indicate biases toward the large-reward choice.

preference, both during motion viewing and around saccade onset (Contra > Ipsi); and (3) more activity when the choice was associated with large reward (red > blue). This neuron's activity thus reflected choice, the strength of motion stimulus leading to the choice, and the reward size expected for the choice. Additional examples are shown in *Figure 2—figure supplement 1*.

Across the sampled population, individual caudate neurons were sensitive to choice, reward context, expected reward size, and motion strength. We measured the selectivity of single-unit activity using multiple linear regression for seven task epochs (*Figure 2C*; *Equation 1*; epochs are defined in

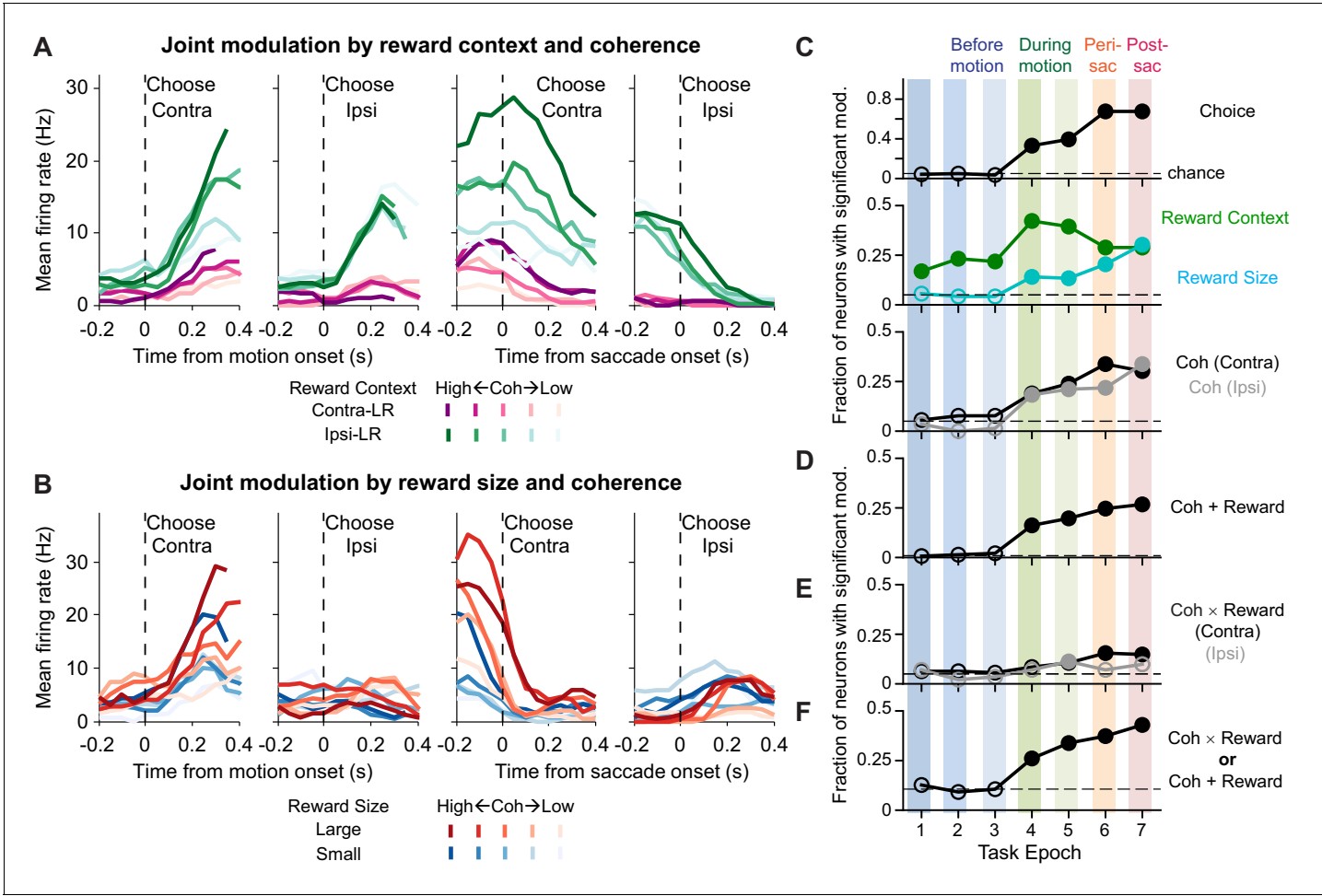

**Figure 2.** Caudate activity reflected motion strength, reward context, choice, and the expected reward size associated with the choice. (A, B) Activity of two example neurons. Shades: coherence levels. Colors: reward context (A) and reward size (B). Firing rates were computed using a 200 ms running window (50 ms steps). Only correct trials were included. (C-F) Fractions of neurons showing significant coefficients for task-related regressors in the seven task epochs defined in *Figure 1A* (see *Equation 1* for the formulation of the regression). Horizontal dashed lines: chance levels, adjusted based on the number of tests used and a 5% chance level for a single test. Filled circles: fractions that were significantly above chance levels (Chi-square test, p<0.05/63 (seven epochs and nine comparisons)). Coh: activity with non-zero coefficients for unsigned coherence values. Coh × Reward: activity with non-zero coefficients for the coherence × reward size interaction. Coh + Reward: activity with non-zero coefficients for coherence on trials with either choice and non-zero coefficients for either reward context or reward size.

The online version of this article includes the following source data and figure supplement(s) for figure 2:

**Source data 1.** Regression results for *Figure 2C–F*, using *Equation 1*.

**Figure supplement 1.** Example neurons with different kinds of task-relevant modulations.

**Figure supplement 2.** Modulation patterns of "combination neurons" during motion viewing.

*Figure 1A*). Selectivity for reward context persisted throughout the trial (green), whereas selectivity for choice, reward size, and motion strength emerged during motion viewing (Epochs 4–7). The selectivity for reward size and motion strength was more prevalent in Epoch 5 (variable duration covering the whole motion viewing period) than in Epoch 4 (fixed duration covering only the early motion viewing period), consistent with a developing latent decision variable that accumulates evidence over time, increasing sensitivity of the regression analysis with longer analysis windows, and/or additional sensitivity to RT closer to the time of saccade. Many neurons also showed joint modulations by both motion strength and either the reward context or expected reward size ("Coh + Reward", *Figure 2D*). A smaller proportion of neurons showed a different form of joint modulation by visual and reward-related information: their activity was sensitive to the interaction between motion strength and reward size ("Coh ×Reward", *Figure 2E*). Overall, 101 out of 142 neurons

showed at least one of these forms of joint modulation in at least one epoch. This fraction was significantly above chance level, even considering the multiple tests done in seven epochs (*Figure 2F*; Chi-squared test, p=0.0035). These neurons also showed heterogenous modulation patterns, as illustrated for the 44 neurons with joint modulation during motion viewing (*Figure 2—figure supplement 2*).

Thus, information about both the visual motion evidence and reward expectation were represented in the caudate nucleus at both the population and single-neuron levels. The combined evidence-reward representations at the single-neuron level in some caudate neurons suggest that the caudate nucleus may contribute directly to the process of integrating both sources of information into the decision.

## Caudate microstimulation evoked reward context-dependent effects on behavior

We next tested if and how the caudate nucleus contributes causally to decisions that balance visual evidence and reward information. We identified caudate sites with task-modulated activity and delivered electrical microstimulation during motion viewing at these sites in randomly interleaved trials. We were particularly interested in microstimulation effects that depended on visual and reward information interactively, which would imply that the caudate nucleus contributes to the integration of those sources of information into the decision.

Consistent with the heterogeneity of response properties identified above and our previous findings (*Ding and Gold, 2012b*), we found examples of microstimulation effects that were either independent of or dependent on reward context. *Figure 3A* shows an example session in which the microstimulation effects depended on visual evidence but not reward context. At this site, microstimulation induced contralateral biases (a leftward shift in the psychometric function), reduced sensitivity to the visual motion evidence (a reduction in the slope of the psychometric function), decreased mean RT for the contralateral choice, and increased mean RT for the ipsilateral choice, with similar magnitudes of changes for the two reward contexts. In contrast, *Figure 3B* shows an example session in which the microstimulation effects depended on both visual evidence and reward context. At this site, microstimulation induced ipsilateral choice biases, slope reduction, and decreases in RT for ipsilateral saccades. The effects were more prominent when the ipsilateral choice was paired with large reward (open circles). Thus, microstimulation changed the relationship between choice/RT and visual evidence in a reward context-dependent fashion. These changes were consistent with a perturbation of a process that incorporates both visual evidence and reward asymmetry information into the decision.

Across the 55 sites tested (*n* = 24 sessions for monkey C, 31 for monkey F), we found many sites with microstimulation effects that reflected different combinations of effects that were independent of reward context and others that showed interactive effects on visual and reward processing. To quantify these diverse microstimulation effects, we fitted logistic (*Equation 2*) and linear (*Equation 3*) functions to the choice and RT data, respectively. We parsed the microstimulation effects into two types. First, the "estim" type measured the average microstimulation effect, independent of reward context (*Figure 3C*). These effects tended to include positive changes in choice bias (biasing toward contralateral choices), negative changes in the slope of the psychometric function (reductions in sensitivity to the visual evidence), negative changes in the intercept of RT, and positive changes in the slope of the RT-coherence function (faster RTs that were less sensitive to motion strength). Second, the "rew × estim" type measured the extent to which the microstimulation effect differed between the two reward contexts (i.e., the interaction between reward context and microstimulation effects), which is the focus of the present study and is equivalent to measuring the microstimulation effects on the monkeys' behavioral adjustments to the asymmetric reward contexts (*Figure 3D*). These effects tended to include positive values for choice bias, negative values for the slope of the psychometric function, negative values for the ipsilateral RT, and positive values for the slope of the RT-coherence function. Values for individual sessions are plotted in *Figure 3—figure supplement 1B*, and summary statistics are in *Supplementary file 1a*. The median "rew × estim"-type effects tended to have the same signs as the corresponding median "estim"-type effects (compare the vertical lines in *Figure 3C and D*), indicating that the microstimulation effects tended to be larger when the large reward was paired with choices contralateral to the site of microstimulation. Note that for the directional choice bias, this tendency may manifest itself as microstimulation inducing either larger

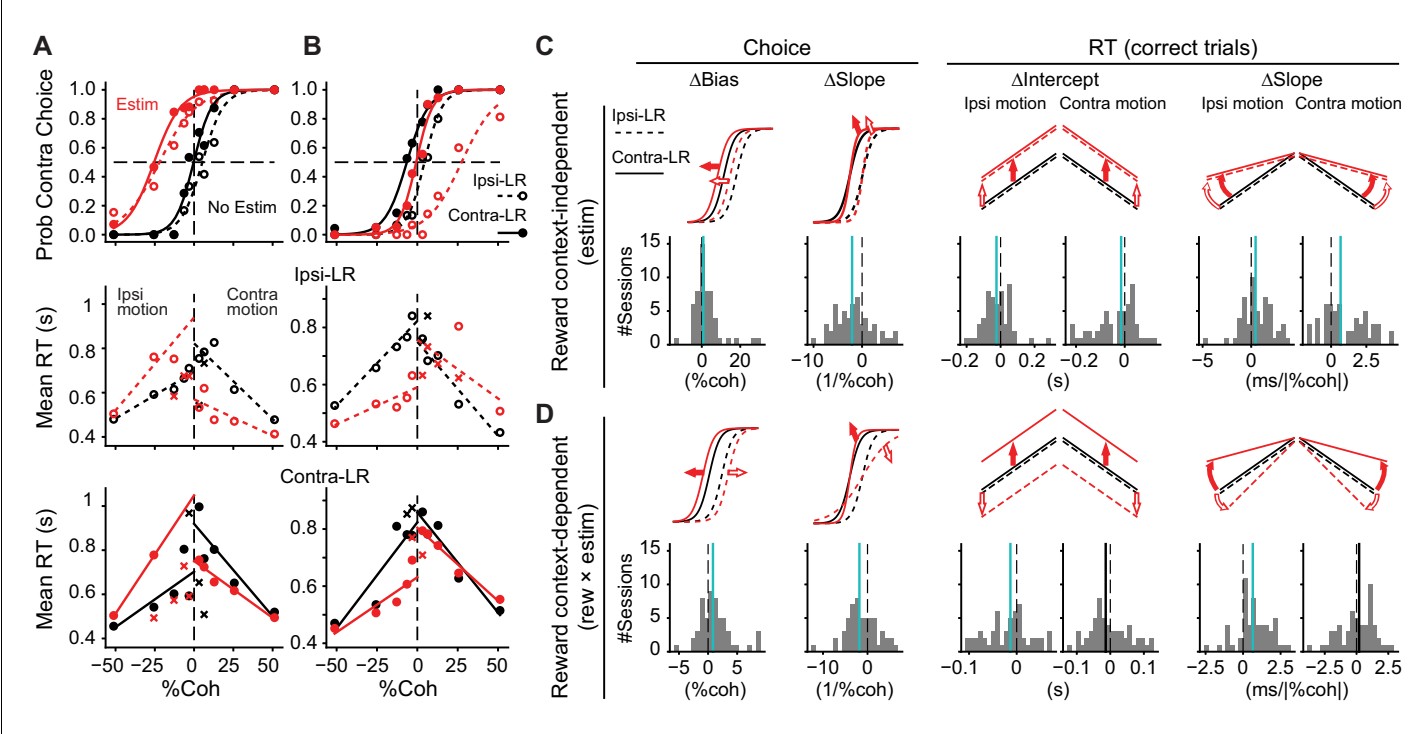

**Figure 3.** Caudate microstimulation affected the monkeys' decision behavior. (A, B) Two example sessions from monkey C showing different patterns of microstimulation effects. Black: trials without microstimulation; red: trials with microstimulation. Open circles and dashed lines: data and fits for blocks in which the ipsilateral choice was paired with large reward and the contralateral choice was paired with small reward. Filled circles and solid lines: data and fits for blocks in which the ipsilateral choice was paired with small reward and the contralateral choice was paired with large reward. Abscissa: signed coherence (positive: motion was toward the contralateral target; negative: motion was toward the ipsilateral target). Top panels: probability of making the contralateral choice. Lines are logistic fits to the choice data. Middle and bottom panels: mean reaction time (RT) for the two reward contexts. Circles and crosses represent correct and error choices, respectively. Lines: linear fits to single-trial RT data. (C, D) Summary of effects induced by microstimulation for both reward contexts (C, estim terms) and by interactions between microstimulation and reward context (D, rew × estim terms). The top rows illustrate the effects on psychometric and chronometric curves for positive values of the corresponding terms. Bottom rows show the histograms of changes in choice bias (logistic shift in %coh), perceptual sensitivity (logistic slope in 1/%coh), and intercept and slope of linear regression of RT as a function of unsigned motion coherence. Thick vertical lines indicate the median values. Green lines, Wilcoxon signed-rank test, p<0.05.

The online version of this article includes the following source data and figure supplement(s) for figure 3:

**Figure supplement 1.** Effects of caudate microstimulation for all sessions.

**Figure 3—source data 1.** Fitting results for choice (logistic) and RT (linear) data.

**Figure supplement 2.** The average RT difference between the two reward contexts alone could not account for the reward context-dependent microstimulation effects.

contralateral biases in "Contra-LR" blocks, or larger ipsilateral bias in "Ipsi-LR" blocks (*Figure 3B*). These dependencies on reward context were similarly apparent when considering the two reward contexts separately (*Figure 3—figure supplement 1A*).

Overall, microstimulation induced at least one statistically reliable effect on choice or RT at 48 of 55 caudate sites (colored dots in *Figure 3—figure supplement 1B*; significantly above the chance level that was adjusted for multiple comparisons, Chi-squared test, p<1e-5). Of these, 27 sites showed at least one "rew × estim"-type effect on choice and/or RT. Most of these sites included effects on psychometric bias, psychometric slope, and/or the slope of RT as a function of motion strength, all of which (unlike the intercept of the RT function) measured the dependence of the monkeys' behavior on motion evidence.

Because we delivered microstimulation throughout motion viewing for an RT task, the difference in average RT between reward contexts may contribute to "rew × estim"-type microstimulation effects. We found that the average RT tended to be longer for the "Ipsi-LR" and "Contra-LR" blocks for the monkeys C and F, respectively (*Figure 3—figure supplement 2A*). However, the difference

in average RT between reward contexts alone cannot explain the "rew $\times$ estim"-type microstimulation effects. For each effect in *Figure 3D*, we performed a linear regression using the difference in RT as the independent variable (*Figure 3—figure supplement 2B*). The percentage of variance explained was lower than 30% for monkey C and lower than 10% for monkey F. Even for the case with the largest explained variance, the relationship was opposite to what would be expected, that is, sessions with larger RT differences tended to show smaller "rew $\times$ estim"-type microstimulation effects on choice bias for monkey C. Thus, the dependence of the microstimulation effects on reward context more likely reflected a causal involvement of the caudate nucleus in balancing visual evidence and reward asymmetry information.

## Microstimulation caused coordinated adjustments to reward-dependent decision biases that mimicked the monkeys' voluntary strategy

We next examined more closely how these diverse microstimulation effects on choice and RT were related to the specific strategies that the monkeys were using to balance visual evidence and reward asymmetry information. We previously showed that, on the same task without microstimulation, the monkeys' choice and RT behaviors can be well described by a drift-diffusion model (DDM), in which noisy visual evidence is accumulated over time until reaching a pre-defined, time-varying, bound (*Figure 4A*; *Fan et al., 2018*; *Ratcliff and Rouder, 1998*; *Zylberberg et al., 2016*). Within this framework, the balance between visual evidence and reward asymmetry is captured by two reward context-dependent parameters that induce asymmetries in the drift rates and the relative bound heights for the two choices, respectively. A change in the drift rate was implemented as additional momentary evidence (*me*) that is added to the motion evidence at each accumulating step. A change in the relative bound heights was implemented as a shift in the starting point (*z*) of the DDM. A positive difference in Δdrift or Δbound between reward contexts, that is, positive Δdrift (rew) or Δbound (rew), leads to more and faster choices to the large-reward option. Either Δdrift (rew) or Δbound (rew) can have similar effects on the psychometric function but are distinguishable via different effects on the chronometric function (see Figure 3—figure supplement 1 in *Fan et al., 2018*). Here, we used the DDM to compare the monkeys' voluntary decision strategies with the effects of microstimulation on those strategies.

As we reported previously, the reward function (average reward rate) on our task depended on perceptual sensitivity, reward context, and the relative sizes of large and small rewards in a given session. The monkeys tended to use positive Δdrift (rew) values that changed in magnitude according to session-by-session variations in the reward function (see Figure 6 in *Fan et al., 2018*), implying fast adaptation within a session. In addition to these positive, but variable, Δdrift (rew) values, the monkeys tended to use negative Δbound (rew) values that also changed systematically with changes in the reward function, such that the values of Δdrift (rew) and Δbound (rew) were negatively correlated with each other across sessions (*Figure 5B*). These coordinated adjustments produced overall biases toward the large-reward choice consistently across sessions (see also *Figure 1C*). We also showed previously that this negative correlation followed the predicted relationship based on a heuristic decision strategy that used features of the session-specific reward function to achieve near-maximal, "good-enough" reward rate (*Fan et al., 2018*).

Based on these findings, we predicted that if the caudate nucleus contributes causally to the implementation of the monkeys' decision strategies, then the effects of caudate microstimulation on Δdrift (rew) and Δbound (rew) would show a similar negative correlation across sessions (caudate sites). Alternatively, if the caudate nucleus contributes causally only to the implementation of specific Δdrift (rew) and/or Δbound (rew) values, we predicted that microstimulation would likely induce uncorrelated changes across sessions (caudate sites). To guide our examination of the experimental data, we considered several likely scenarios of microstimulation effects (*Figure 4C–F*; see Materials and methods for simulation parameters). For all scenarios, we assumed that, in the absence of microstimulation: 1) the monkeys used negatively correlated Δdrift (rew) and Δbound (rew) values across sessions (*Figure 4B*); and 2) Δdrift (rew) and Δbound (rew) were combined with variable baseline values (drift0 and bound0) to generate the observed choices and RTs for each reward context (*Figure 4C–F*, left). In the first scenario, we assumed that the caudate nucleus is not involved in the coordination and contributes only to the implementation of reward context-independent *me* and *z* adjustments, that is, microstimulation is expected to affect the baseline drift0 and bound0 independently (*Figure 4C*). In the second scenario, we again assumed that the caudate

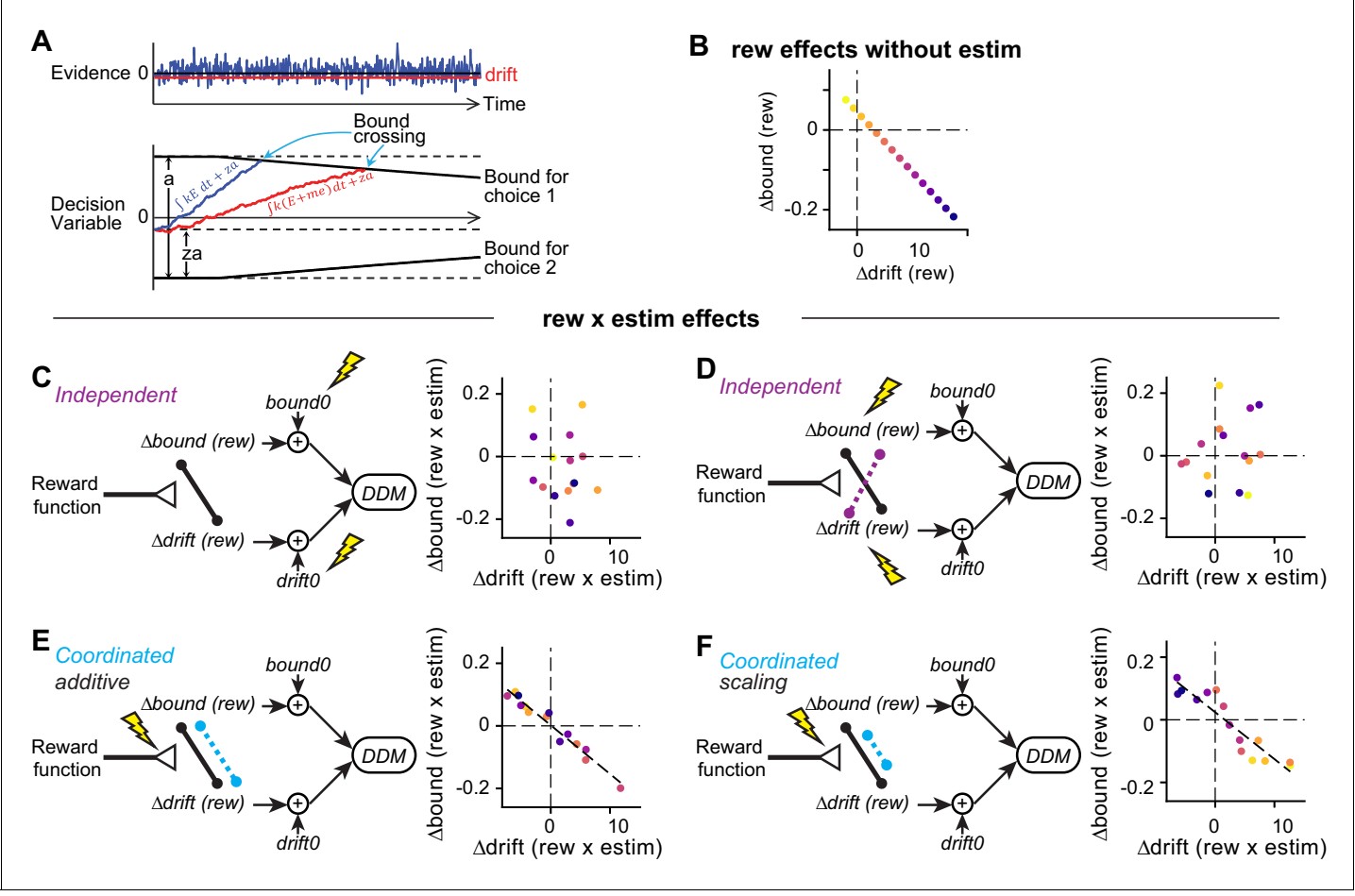

**Figure 4.** Illustration of correlation patterns induced by different hypothesized actions of microstimulation. (**A**) Drift-diffusion model. Motion evidence (*E*) is modeled as samples from a Gaussian distribution (mean = signed coherence, variance = 1). The decision variable is computed as the time integral of *E* and compared at each time point to two (possibly time-varying) decision bounds. Crossing of either bound results in the corresponding choice. RT is modeled as the sum of the time to bound crossing and a non-decision time. Bias can be induced with offsets in evidence (*me*, biasing the drift rate) or relative bound heights (*z*, biasing bounds). (**B**) For the simulations in (**C-F**), reward context-dependent modulation of drift and bound on trials without microstimulation (i.e. Δdrift (rew) and Δbound (rew)) were negatively correlated across sessions. Each colored circle represents a session. (**C and D**) Microstimulation induces independent changes in baseline drift0 and bound0 values (**C**) or independent changes in Δdrift (rew) and Δbound (rew) (**D**). Note the absence of negative correlation in the rew × estim effects. (**E**) Microstimulation additively affected reward context-dependent adjustments in drift and bound in a coordinated manner. The dashed line represents a significant correlation. (**F**) Microstimulation additively and multiplicatively affected reward context-dependent adjustments in drift and bound in a coordinated manner. Note that the rew × estim effects were negatively correlated between drift and bound (dashed line). The rew × estim effects were negatively correlated with the reward effects without microstimulation in F (compare the orders of color progression to that in **B**), but not (**C-E**). See Materials and methods for simulation parameters.

nucleus is not involved in the coordination, but in this case contributes to the implementation of reward context-dependent *me* and *z* adjustments; i.e., microstimulation is expected to affect Δdrift (rew) and Δbound (rew) independently (*Figure 4D*). In the third and fourth scenarios, the caudate nucleus contributes to the coordination process. Microstimulation added correlated changes in Δdrift (rew) and Δbound (rew) (*Figure 4E*) and further scaled the correlated Δdrift (rew) and Δbound (rew) values (*Figure 4F*). Notably, only the coordinated-effect scenarios (*Figure 4E and F*) predict a similar relationship for interaction effects between reward and microstimulation. Moreover, only the coordinated-scaling scenario (*Figure 4F*) predicts that the "rew × estim" effects should vary systematically with "rew" effects without microstimulation across sessions (compare the color progression in *Figure 4B and C–F*).

We fitted the monkeys' single-trial data with different DDM variants (see fits to example sessions in *Figure 5—figure supplement 1* and summary data in *Figure 5—figure supplement 2*). For the

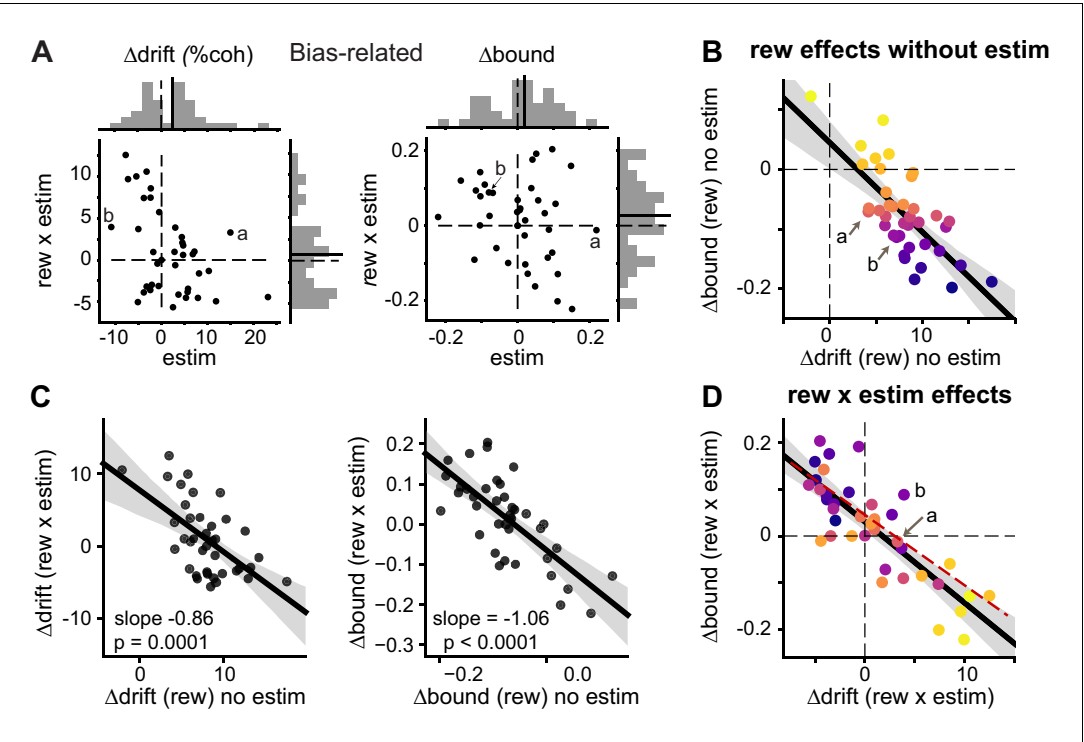

**Figure 5.** Microstimulation induced correlated changes in the reward modulation of drift and bound. (**A**) Scatter plots of changes in drift and bound induced by electrical microstimulation (abscissa and top histograms) and by interactions between electrical microstimulation and reward condition (ordinate and right histograms). Solid lines in histograms: mean values across sessions, *t*-test, p>0.05. Labels (a, b) correspond to the example sessions in *Figure 3A and B*, respectively. (**B**) On trials without microstimulation, the differences in Δdrift and Δbound between the two reward contexts were negatively correlated. Line and shaded area: linear regression and 95% confidence interval, *t*-test, p<0.0001. "a" and "b" indicate the data points for the examples in *Figure 3A and B*. Data are color-coded by the values of Δbound (rew) no estim. (**C**) Scatter plots of reward effects on trials without microstimulation (abscissa) and interaction effects (ordinate) for Δdrift (left) and Δbound (right). Lines represent results of linear regression (shaded area: 95% confidence interval). (**D**) The interaction effects (Δdrift (rew x estim) and Δbound (rew x estim)), equivalent to the difference between "Δdrift/bound (rew) with estim" and "Δdrift/bound (rew) no estim", were negatively correlated. Same format as B. *t*-test, p<0.0001. Red dashed line re-plots the linear regression results from *Figure 5B*, using the appropriate range of Δdrift (rew x estim) as x-values. Data are color-coded by the values of Δbound (rew) no estim. Note that the roughly reversed orders of color progressions in B and D is most consistent with simulated effects in *Figure 4F*.

The online version of this article includes the following source data and figure supplement(s) for figure 5:

**Source data 1.** Fitting results for choice and RT data using the DDM for sessions with significant microstimulation effects.

**Figure supplement 1.** DDM fits to example sessions in *Figure 3*.

**Figure supplement 2.** DDM fitting results.

**Figure supplement 3.** Biases in drift and bounds together accounted for biases measured in logistic fits.

**Figure supplement 4.** Both monkeys showed similar correlation patterns between Δdrift and Δbound.

remaining analyses, we focused on 39 sessions in which microstimulation affected at least one DDM component. In these sessions, the full DDM model, in which all parameters were allowed to change with reward context and microstimulation, considerably outperformed a reduced model ("No Estim"), in which all parameters were allowed to change with reward context but not microstimulation (*Figure 5—figure supplement 2A*). In other words, if the AIC value for the full model was less than that for the "No Estim" model by a difference of at least 7 (to the left of the red arrow in *Figure 5—figure supplement 2A*), we assumed that a change in at least one DDM parameter was required to account for microstimulation effects in that session. The reduced model without microstimulation effects on parameters describing the collapsing bound ("NoCollapse") outperformed the

full model in most sessions, indicating that caudate microstimulation did not affect the time course of the bound height (*Figure 5—figure supplement 2B*). In contrast, the full model tended to outperform other reduced models, indicating that caudate microstimulation evoked changes in parameters controlling the speed-accuracy tradeoff (*a* and *k*), biases in choice and RT (*me* and *z*), and non-decision times (*t_contra* and *t_ipsi*; *Figure 5—figure supplement 2C*). Consistent with the results from logistic fits, the microstimulation effects on these parameters were often reward context-dependent (*Figure 5—figure supplement 2D*; many data points deviated from the unity-slope lines). Summary statistics of the microstimulation effects, partitioned into "estim" and "rew × estim" types, are shown in *Supplementary file 1b*.

We found that the microstimulation effects on Δdrift (rew) and Δbound (rew) were most consistent with the coordinated-scaling scenario in *Figure 4F*, based on several observations. First, DDM variants with microstimulation-induced changes in both *me* and *z* outperformed variants with no such changes in *me* or *z* (*Figure 5—figure supplement 2B and C*; the "NoME" and "NoZ" models provided the best fits for only three sessions), suggesting that caudate microstimulation affected both drift rates and bounds. Second, despite the variations in the microstimulation effects on *me* and *z* across sessions (*Figure 5A*), the combined effects produced similar changes in bias to those measured using logistic fits (*Figure 5—figure supplement 3*), reminiscent of our previous observation that the monkeys used both Δdrift (rew) and Δbound (rew) to produce consistent reward biases. Third, the interaction ("rew × estim") effects on Δdrift and Δbound maintained a highly consistent relationship across sessions. As expected for a coordinated bias strategy, the monkeys' voluntary, reward-dependent adjustments were negatively correlated on trials without microstimulation (*Figure 5B*). Strikingly, and consistent with only the coordinated-effects scenarios in *Figure 4E and F*, the interaction effects between reward and microstimulation followed almost the same negative correlation (*Figure 5D*, compare red dashed line and black solid lines). The same patterns were observed in both monkeys (*Figure 5—figure supplement 4*). Fourth, there were strong negative correlations between the reward effects on trials without microstimulation and interaction effects for both Δdrift and Δbound (*Figure 5C*; also compare the orders of color progressions in B and D). This result is consistent with the scenario in *Figure 4F* and thus suggests the presence of a multiplicative component in the microstimulation effects on coordination. Together, these results show that caudate microstimulation recapitulated the coupling between reward context-dependent adjustments to drift rates and relative bound heights that was evident in the monkeys' voluntary adjustments.

We considered four alternative explanations and found that the correlated "rew × estim" effects on drift and bound were not artifact, trivial, or a necessary consequence of reward-dependent biases. First, our methods of DDM fitting and parsing reward- and microstimulation-related effects might have introduced artificial correlations. This explanation would predict a similar correlation in the overall, reward-independent microstimulation effects on drift and bound. We did not observe such a correlation between Δdrift (estim) and Δbound (estim), despite similar magnitudes of changes to their reward-dependent counterparts (*Figure 6A*). To further control for this scenario, we shuffled all the fitted DDM parameters across sessions and simulated a new data set with matched trial numbers for each session ("Shuffle 1"). We fitted the full DDM to these simulated data and did not observe a negative correlation as in the original data (*Figure 6B–D*). Fits with the "NoCollapse" variant did not show negative correlation, either (*Figure 6—figure supplement 1*).

Second, the coordinated effects of microstimulation on Δdrift and Δbound might have been a trivial consequence of reward-dependent biases, independent of the reward context-dependent coordination of the two quantities. This possibility is countered by three findings: 1) the presence of correlated reward-dependent adjustments alone do not necessarily produce correlated "rew × estim" effects (e.g. *Figure 4C and D*); 2) removing coordinated, reward-induced effects by shuffling the best-fitting values of those two parameters independently across sessions (as in *Figure 4D*) resulted in substantially less correlated "rew × estim" effects than in the real data (*Figure 6E*, "Shuffle 2" and "Shuffle 3", red and gray solid lines, compare to the data in black). That is, even with the mean and variance values matched exactly to the experimental data, the model in *Figure 4D* cannot capture the correlated pattern we observed in the experimental data; and 3) partial shuffling that disrupted only possible relationships with session-specific properties (e.g. microstimulation sites or voluntary performance), by shuffling the paired Δdrift and Δbound across sessions, also significantly weakened the correlation between "rew × estim" effects (*Figure 6E*, "Shuffle 4").

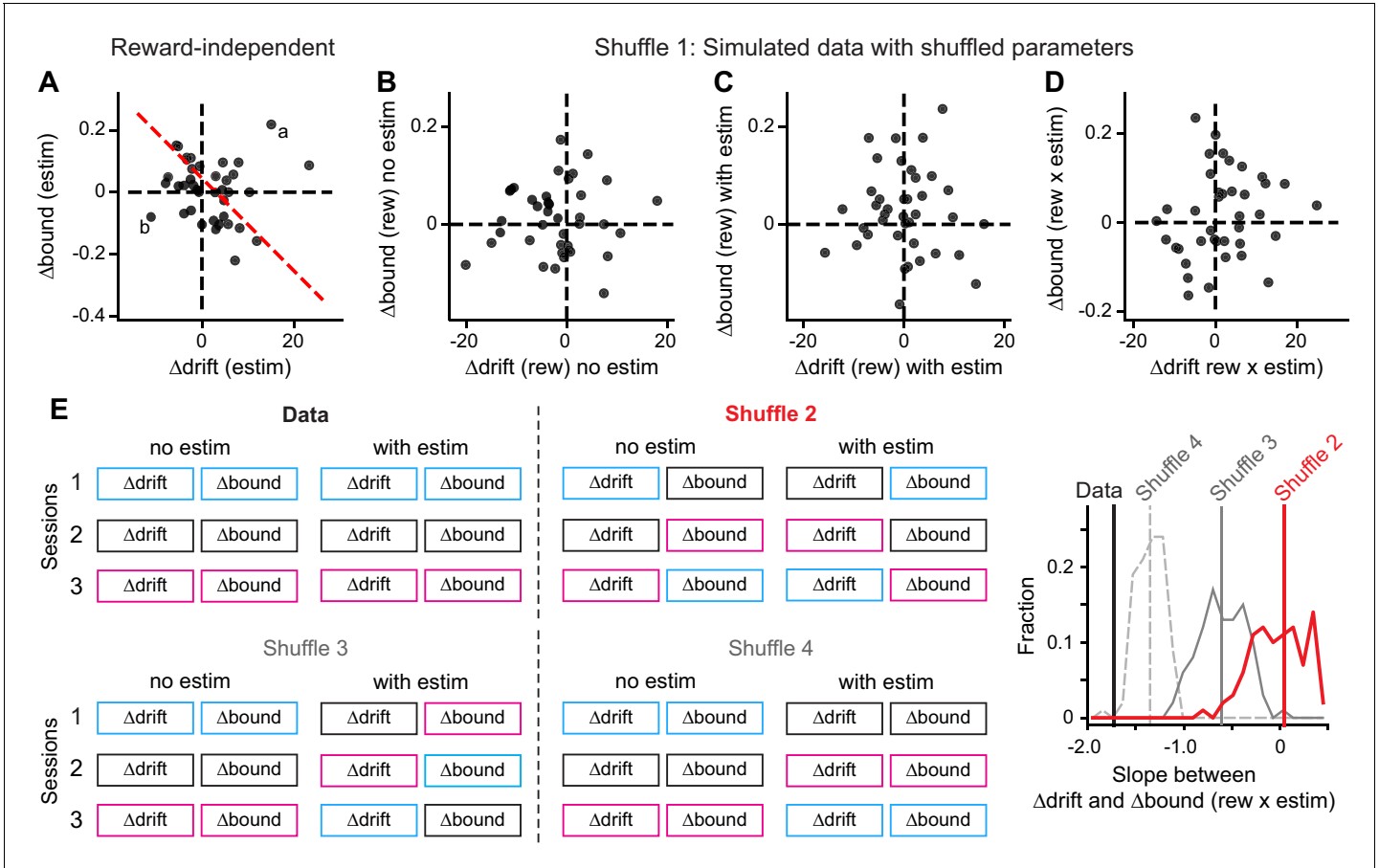

**Figure 6.** Control analysis results. (**A**) The average microstimulation effects on Δdrift and Δbound that were independent of reward context were not correlated. Same format as *Figure 5D*. Linear regression, *t*-test, p=0.60. (**B-D**) The patterns of correlation in *Figure 5B and D* were not observed in data simulated with shuffled DDM parameters (Shuffle 1). Δdrift and Δbound values were obtained by fitting the simulated data with the "Full" DDM model. Linear regression, *t*-test, p=0.78, 0.81, 0.26 for the three panels, respectively. (**E**) The observed relationship between Δdrift (rew × estim) and Δbound (rew × estim) was different from those observed in shuffled data. Shuffle 2 (red): shuffle Δdrift and Δbound values for trials with and without microstimulation across sessions independently. Shuffle 3 (gray, solid): shuffle Δdrift and Δbound values for microstimulation trials only and across sessions independently. Shuffle 4 (gray, dashed): shuffle Δdrift and Δbound values for microstimulation trials only across sessions while maintaining the coupling between drift and bound. Histograms were obtained from 200 shuffles for each method. Vertical lines indicate the mean slope values. The online version of this article includes the following figure supplement(s) for figure 6:

**Figure supplement 1.** Δdrift and Δbound values obtained by fitting the simulated data to the "NoCollapse" DDM model did not show the same correlation patterns as the original data.

**Figure supplement 2.** Both monkeys showed significant negative correlation between Δdrift (rew) and Δbound (rew) on trials before microstimulation began.

**Figure supplement 3.** The correlated Δdrift (rew × estim) and Δbound (rew × estim) effects represented the dominant features of the data.

Third, it is possible that the shared negative correlations in *Figure 5B and D* resulted from persistent effects of microstimulation throughout each session, that is, a spill-over of microstimulation effect. This possibility is inconsistent with three observations: 1) the negative correlations were present between Δdrift (rew) and Δbound (rew) in sessions without microstimulation (data not shown; *Fan et al., 2018*); 2) As described in Methods, microstimulation sessions began with trials for unit recording only, before any microstimulation was delivered. During these recording-only trials, negative correlations between Δdrift (rew) and Δbound (rew) were also observed, similar to the correlations during later, no-microstimulation trials interleaved with microstimulation (*Figure 6—figure supplement 2*), indicating that microstimulation was not necessary for the observed negative correlations; and 3) the reward effects on trials without microstimulation were

negatively correlated with the interaction effects (*Figure 5C*), instead of a positive dependency as predicted by a "spill-over" mechanism with linear or exponential decay.

Fourth, given the inter-session variability in microstimulation effects, it is possible that the negatively correlated Δdrift (rew × estim) and Δbound (rew × estim) reflected only a minor consequence of caudate microstimulation. We examined this possibility by comparing: 1) the first principal component (PC) for the eight fitted parameters ([*me* or *z*] × [contra-LR or ipsi-LR] × [estim on or off]), with 2) the first PC for the two parameters Δdrift (rew × estim) and Δbound (rew × estim). We projected the eight-parameter and two-parameter data to these two PCs, respectively. The projections showed a strong correlation (*Figure 6—figure supplement 3*; rho = 0.92, p=1.3e-16), indicating that the coordinated Δdrift (rew × estim) and Δbound (rew × estim) values reflected the dominant feature of the experimental data; that is, the dominant effect of caudate microstimulation on reward-related biases.

Taken together, these results suggest that the shared correlation patterns in reward context-dependent voluntary and evoked adjustments were not trivial, but instead reflected the caudate nucleus' causal involvements in mediating the monkeys' specific strategy to balance visual evidence and reward asymmetry information.

## Coordinated microstimulation effects depended on neural selectivity patterns and baseline adjustments

The microstimulation effects on Δdrift (rew) and Δbound (rew) tended to be negatively correlated with each other but also showed substantial variability across sessions (*Figure 5D*). We examined three potential sources of these effects: 1) the monkey's baseline (microstimulation-independent) adjustments in Δdrift (rew) and Δbound (rew) (*Figure 5B*), 2) the task-related activity patterns of neurons recorded near the stimulation site, and 3) the anatomical location of the stimulation site. As detailed below, we found that microstimulation effects on Δdrift (rew) and Δbound (rew) depended systematically on both the baseline adjustments and neural selectivity at the site of microstimulation, but not the anatomical location of the site.

We first performed multiple linear regressions, using the baseline adjustments and neural selectivity at the site of microstimulation as the independent variables and the microstimulation effects as the dependent variables (*Equation 4*). We quantified the baseline adjustments in terms of the two principal components (PCs) of the relationship between Δdrift (rew) and Δbound (rew), which to a first approximation divides the effects into the average coordinated trajectory (the projection onto PC1) and the deviation from the average trajectory (the projection onto PC2). We quantified neural selectivity during motion viewing (Epoch 5in *Figure 1A*, when the decisions were formed) for each site using a multiple linear regression that also allowed us to identify neurons with combined modulations by visual evidence and reward context (*n* = 16 sites total, 9 and 7 for monkeys C and F, respectively; *Equation 1*). We quantified the microstimulation effects as the projections of Δdrift (rew × estim) and Δbound (rew × estim) onto the PCs defined from the baseline adjustments, which measured the degree to which the "rew × estim" interaction effects conformed to the voluntary adjustments in response to the asymmetric reward contexts (*Figure 7A and B*, right panels).

We found that the microstimulation effects along the average coordinated trajectory (i.e. the PC1 projection of rew × estim effects) depended: 1) negatively on the baseline effects along that trajectory (i.e., the PC1 projection of baseline Δdrift (rew) and Δbound (rew)), 2) negatively on neural selectivity for choice at the site of microstimulation, 3) positively on neural selectivity for reward context at the site of microstimulation, and 4) positively on neural selectivity for motion coherence for trials with contralateral choices at the site of microstimulation (*Figure 7C*, colored boxes). The explained variance of this regression was 89% and significantly higher than that of shuffled data (*Figure 7D*, top panel). The microstimulation effects perpendicular to the average coordinated trajectory also depended positively on neural selectivity for motion coherence for trials with contralateral choices, but the explained variance was not significantly different from that of shuffled data. In contrast, for other sites without combined modulations by motion coherence and reward context (*n* = 23 sites total, 9 and 14 for monkeys C and F, respectively), the microstimulation effects along the average coordinated trajectory also depended negatively on the baseline adjustments along that trajectory, but the microstimulation effects did not otherwise depend on the baseline adjustments or neural tuning from that session (*Figure 7C*, bottom panel). For these sites, the explained variance was substantially lower than for the sites with combined modulations and did not differ from the shuffled

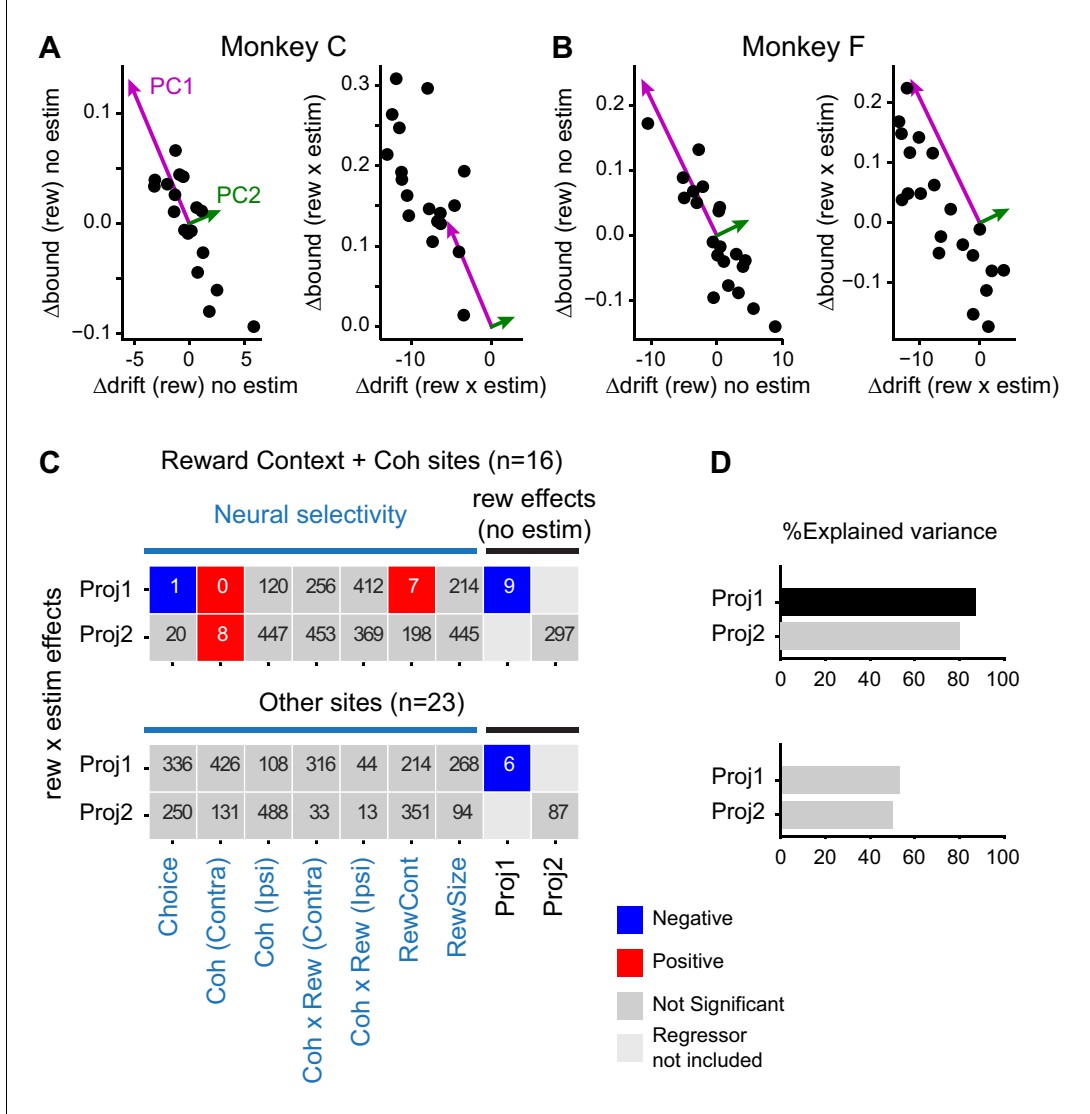

**Figure 7.** Microstimulation effects on coordination depended on neural selectivity and baseline adjustments of drift and bound. (**A and B**) Principal components (PCs) were estimated for reward context modulation of Δdrift and Δbound without microstimulation (left panels; mean subtracted) for the two monkeys separately. The values of Δdrift (rew × estim) and Δbound (rew × estim) were projected onto these two PCs (right panels). (**C**) Visualization of the dependence of rew × estim effects on neural selectivity and reward effects without microstimulation for sites with combined modulations by motion coherence and reward context (top) and for other sites (bottom). Proj1 (2): projection onto PC1(2). Each row shows the results from one multiple linear regression, with the quantity on the ordinate as the dependent variable and the quantities on the abscissa as the independent variables. Significance was assessed with bootstrap methods using 1000 shuffles of the independent variables. The number in each box indicates the number of shuffles with the same or stronger effect as the experimental data (e.g., six means an estimated p value of 0.006 and 0 means an estimated p value of < 0.001). Colors indicate the signs of regression coefficients (significance criterion: p<0.0125 = 0.05/4-regressions). (**D**) Percent explained variance for the regressions in (**C**), respectively. A black bar indicates that the experimental value is outside the 95 percentile of the bootstrapped distribution.

The online version of this article includes the following source data and figure supplement(s) for figure 7:

**Source data 1.** Neural selectivity for different task factors at the microstimulation sites.

**Figure supplement 1.** MRI reconstruction of recording and microstimulation sites.

data (*Figure 7D*, bottom panel). These results imply that the microstimulation effects on the monkeys' reward bias strategy reflected both their baseline reward-driven biases and specific contributions of caudate neurons at the site of microstimulation, particularly those encoding both reward context and motion coherence, instead of non-specific microstimulation effects from altering overall caudate activity.

In contrast, the microstimulation effects did not show consistent relationships with the anatomical locations of microstimulation sites. We reconstructed the locations of single-unit recording and microstimulation sites on MRI images based on the track coordinates and electrode depth (*Figure 7—figure supplement 1*). In monkey C, the recording sites spanned from 5 mm anterior to the anterior commissure to 3 mm posterior. In monkey F, the recording sites spanned from the anterior commissure to 7 mm anterior. Neurons with combined modulation by motion and reward-related quantities during motion viewing were observed at all anterior–posterior (A–P) levels (*Figure 7—figure supplement 1A*, red circles; corresponding to neurons identified in *Figure 2—figure supplement 2*). Microstimulation sites were sampled in a smaller region for both monkeys (*Figure 7—figure supplement 1A*, cyan crosses). For our sampled sites, we observed certain tendencies, but these tendencies were not consistent across monkeys. For example, in monkey F, but not monkey C, sites with rew × estim effects on the monkeys' choice and RT, as measured separately using logistic and linear fits, appeared clustered at more ventral locations (*Figure 7—figure supplement 1B and C*). In monkey C, but not monkey F, Δdrift (rew × estim) appeared to be greater at more ventral locations (*Figure 7—figure supplement 1D*). Although we sampled a larger region in monkey C than in monkey F, monkey F showed a larger range of Δdrift (rew × estim) and Δbound (rew × estim) (*Figure 7—figure supplement 1D and E*). These results suggest that anatomical organization patterns are likely variable and confined to small areas in the caudate nucleus. Consistent with the lack of a large-scale organization pattern, adding location data as independent variables in the regressions for *Figure 7* did not significantly improve the explained variance. These results suggest that the rew × estim effects of caudate microstimulation were more closely linked with neural selectivity at the sites than the physical locations of the sites.

## Discussion

For monkeys making saccade decisions based on noisy visual motion evidence and asymmetric reward contexts, we observed that the activity of many single neurons in the caudate nucleus was sensitive to both motion strength and reward context or expected reward size. We further showed that microstimulation at these caudate sites evoked changes in the monkeys' choices and RTs that depended on the reward context and involved adjustments to the drift rates and relative bound heights of an accumulate-to-bound (DDM) decision process. These reward context-dependent microstimulation effects mimicked the coordinated adjustments that the monkeys made, in the absence of microstimulation, to the drift rates and relative bound heights in response to changes in reward context. These evoked, coordinated adjustments also depended on neural selectivity at the microstimulation sites. Taken together, these findings support theoretical proposals that the basal ganglia pathway implements general-purpose computations for action selection, including selection based on multiple types of inputs (*Berns and Sejnowski, 1995*; *Bogacz and Gurney, 2007*; *Mink, 1996*; *Rao, 2010*; *Ratcliff and Frank, 2012*; *Redgrave et al., 1999*). Below we consider how these results inform our understanding of the specific role of the caudate nucleus in forming decisions that integrate sensory and reward information.

The role of the caudate nucleus in coordinating the reward context-dependent modulation of choice bias-related computations (drift rates and bounds in the DDM) appears to be distinct from its previously demonstrated roles in mediating sensory evidence-based decisions. For example, we previously showed that the caudate nucleus contributes causally to adjusting the decision bound ($z$ in the DDM) for decisions based on visual information alone (*Ding and Gold, 2012b*). In the present study, such an effect would be more related to the Δbound (estim) measurement, instead of the "rew × estim" effects. We did not observe a consistent Δbound (estim) effect in the present study, possibly because of differences in task context (the previous study used an equal reward decision task), the targeted neural subpopulation (the previous study targeted sites with choice-selective activity, whereas the present study targeted sites with additional selectivity for reward-related factors), and/or anatomical locations of the stimulation sites (*Figure 7—figure supplement 1A*,

compare blue and cyan crosses; in the previous study, the sites were more posterior and dorsal in monkey C and more anterior and dorsal in monkey F).

The differences in anatomical locations likely also contributed to the apparent sign difference in reward-independent effect on choice bias: microstimulation at sites in our previously published study induced predominantly ipsilateral biases, whereas microstimulation at other caudate sites induced predominantly contralateral biases for the same equal-reward task (Doi et al., Society for Neuroscience 2012, 198.10). In the present study, the reward-independent microstimulation effects on bias were highly variable and showed an average contralateral bias. In comparison, studies of striatal perturbation in rodents showed consistent inactivation-induced ipsilateral biases and activation-induced contralateral biases (*Wang et al., 2018*; *Yartsev et al., 2018*). Interestingly, the effects of caudate microstimulation in monkeys seem more variable than the effects of striatal perturbations in rodents (*Amemori et al., 2018*; *Wang et al., 2018*; *Yartsev et al., 2018*), possibly reflecting the smaller relative volume of tissue that were affected and consequently greater dependence on local activity patterns in the monkey caudate. This difference may also contribute to the lack of effects of striatal lesions on rats' decisions on a task with manipulations of olfactory evidence and reward context (*Wang et al., 2013*), along with other differences between species (e.g. the extent of functional equivalence between rat dorsomedial striatum and monkey caudate nucleus) and task designs (e.g. interleaved trials with and without microstimulation minimized any compensatory effects).

The coordinated microstimulation effects reported here also differ from earlier demonstrations of caudate's involvement in reward/value-dependent behaviors. We found some caudate neurons that exhibited coherence and reward size modulations that were consistent with a flexible value signal that has been observed in the monkey striatum (*Kawagoe et al., 1998*; *Kim and Hikosaka, 2013*; *Samejima et al., 2005*). However, in our task, the coherence-driven decision variable can also be used as a surrogate for confidence, which should be examined in more detail to disentangle its representation from a representation of value in the caudate. One perturbation study showed that pairing caudate microstimulation consistently with a specific stimulus can alter the subjects' choice by increasing the estimated value of that stimulus (*Santacruz et al., 2017*). In our study, such an effect on choice bias would have been canceled when microstimulation was delivered randomly but counterbalanced for the two motion directions. Another study showed that pairing pre-saccade caudate microstimulation with a saccade direction can alter RTs on an instructed saccade task with highly visible stimuli (*Nakamura and Hikosaka, 2006a*). Similar effects likely contributed to our observed changes in non-decision times but do not account for our effects on choice. Finally, caudate microstimulation can prolong saccade RTs on an instructed saccade task by reducing the rate of rise in the LATER model (*Watanabe and Munoz, 2010*), which assumes a linear rise to a fixed bound (*Carpenter and Williams, 1995*; *Reddi et al., 2003*). Similar effects likely contributed to our observed changes in parameter $k$ in the DDM but not the effects on reward-dependent biases that we found. Thus, the coordinated "rew $\times$ estim" effects on drift rates and relative bound heights that we observed here reflect a computational role for the caudate nucleus in reward-biased decisions that is distinct from the roles that have been identified previously.

The caudate nucleus' likely role in coordinating a cognitive process, within the realm of a voluntary decision strategy, is reminiscent of the demonstration that microstimulation in the premotor areas can evoke coordinated body movements, within the realm of natural defensive maneuvers (*Graziano et al., 2002*). Together, these results suggest a general principle of brain organization, such that motor or cognitive primitives are aggregated into behaviorally relevant combinations. Such a hierarchical organization might facilitate learning and flexible adaptation of both motor and cognitive behaviors (*Brooks, 1986*). Within the hierarchy, the coordinated "rew $\times$ estim" effects on drift rates and relative bound heights, which measured changes in the reward-dependent biasing components of the decision process, might reflect the contributions of a subset of caudate neurons to the coordination of cognitive primitives. In contrast, the uncorrelated "estim" effects on drift rates and relative bound heights, which measured changes in reward-independent biases in the decision process, may reflect another subset of caudate neurons' contributions to the implementation of those primitives. These results thus suggest that the caudate nucleus may serve multiple decision-related functions, consistent with the heterogeneous neural modulation patterns across the population (*Nakamura and Ding, 2017*).

Our interpretation of the caudate nucleus' roles was based on an algorithmic description of the decision process using the DDM framework. Given current uncertainty in the field about the neural

implementation of DDM-like decision processes (*Lo and Wang, 2006*; *Rao, 2010*; *Schall, 2019*; *Wei et al., 2015*), more work is needed to understand how the caudate nucleus interacts with the rest of the cortico-basal ganglia circuitry to implement these algorithms for sensory evidence- and reward-dependent decisions. Here, we propose some possibilities for implementing the negatively correlated Δdrift (rew) and Δbound (rew) that we observed, as follows.

First, Δdrift (rew) and Δbound (rew) could be implemented separately in the brain but then converge in the caudate to be coordinated. Δdrift (rew) might reflect reward modulation of the sensory representation in the middle temporal (MT) visual area, where microstimulation can induce choice biases that depend on the average reward expectation for the two choices (*Cicmil et al., 2015*). Such modulation might also be relayed from frontal or parietal regions to the caudate nucleus, although existing data, obtained using a similar task with a fixed motion-viewing duration, argue against the direct involvement by lateral intraparietal area (LIP) neurons in implementing reward-dependent drift rate biases (*Rorie et al., 2010*). In contrast, Δbound (rew) might reflect reward context-dependent activity in the caudate nucleus and elsewhere (including in the LIP), which can emerge before stimulus onset (*Coe et al., 2002*; *Ding and Gold, 2010*; *Ding and Hikosaka, 2006*; *Ikeda and Hikosaka, 2003*; *Kobayashi et al., 2007*; *Lauwereyns et al., 2002a*; *Lauwereyns et al., 2002b*; *Roesch and Olson, 2003*; *Rorie et al., 2010*; *Sato and Hikosaka, 2002*).

Second, Δdrift (rew) and Δbound (rew) could be implemented as a reward context-dependent, time-varying dynamic bias in the decision variable. The idea of a dynamic bias has been proposed to account for the effects of prior information on perceptual decisions (*Hanks et al., 2011*). In our case, the caudate nucleus may control the magnitude and time course of such a bias according to reward context. A similar involvement of the caudate nucleus in the incorporation of prior information may contribute to deficits observed in Parkinsonian patients (*Perugini et al., 2016*).

Third, coordinated Δdrift (rew) and Δbound (rew) might be equivalent to reward context- and time-dependent asymmetric modulations of the two choice bounds. Although caudate activity does not appear to reflect the final bound height at the time of decision (*Ding and Gold, 2012b*), the caudate nucleus may contribute to modulation of the bounds in the earlier phase of decision formation to bias decisions toward the large-reward options. Future experiments with shorter-duration perturbation delivered at different time points during motion viewing may provide further insights into the caudate nucleus' roles (*Yartsev et al., 2018*).

Fourth, Δdrift (rew) and Δbound (rew) might be implemented as reward modulation of network connectivity in a recurrent cortico-basal ganglia network (*Lo and Wang, 2006*; *Wei et al., 2015*). It would be interesting to test if and how perturbations of other components of this network (e.g. the frontal and parietal cortical areas, superior colliculus) affect performance on our task.

Collectively, our results suggest that caudate neurons contribute causally to decisions that combine asymmetric rewards and visual information. These contributions are diverse and complex, possibly operating on different levels in hierarchical cognitive processes. Such diversity of functions might partially explain the complicated nature of decision-making impairments with striatal dysfunction.

# Materials and methods

**Key resources table**

| Reagent type (species) or resource | Designation | Source or reference | Identifiers | Additional information |
|---|---|---|---|---|
| Software and Algorithms | Python 3.5 | Python Software Foundation | https://www.python.org/ | |
| Software and Algorithms | MATLAB | Mathworks | https://www.mathworks.com | |
| Software and Algorithms | Psychophysics Toolbox | *Kleiner et al., 2007* | http://psychtoolbox.org/ | |
| Software and Algorithms | Pandas v0.19.2 | Python Data Analysis Library | https://pandas.pydata.org/ | |

*Continued on next page*

*Continued*

| Reagent type (species) or resource | Designation | Source or reference | Identifiers | Additional information |
|---|---|---|---|---|
| Software and Algorithms | Scikit-learn v0.18.1 | *Pedregosa et al., 2011* | https://scikit-learn.org/stable/ | |
| Software and Algorithms | Scipy v0.18.1 | SciPy.org | https://docs.scipy.org/doc/scipy/reference/stats.html | |

## Experimental model and subject details

We used two adult male rhesus macaques (*Macaca mulatta*) for this study. They were first trained extensively on an equal-reward reaction-time random-dot motion discrimination task (*Ding and Gold, 2010*; *Ding and Gold, 2012a*; *Ding and Gold, 2012b*) and then trained with the asymmetric-reward contexts (*Fan et al., 2018*). All training and experimental procedures were in accordance with the National Institutes of Health Guide for the Care and Use of Laboratory Animals and were approved by the University of Pennsylvania Institutional Animal Care and Use Committee (protocol #804726).

## Method details

### Behavioral task

Task details are reported elsewhere (*Fan et al., 2018*). Briefly, a trial began with a central fixation point presentation (*Figure 1A*). Upon acquiring and maintaining fixation, two choice targets were presented to inform the monkeys the two possible motion directions. After a random delay picked from a truncated exponential distribution (mean = 0.7 s, range: 0.4–2.5 s), the fixation point was dimmed and a random-dot kinematogram was shown at the center of the screen ("motion onset"). For each trial, the kinematogram had a constant speed of 6°/s, aperture size of 5°, and randomly interleaved motion direction and strength (five levels of coherence: 3.2, 6.4, 12.8, 25.6, 51.2%). The monkey reported the perceived motion direction by making a self-timed saccade to the corresponding choice target. A minimum 50 ms latency was imposed, although the monkeys rarely made fast-guess responses during this study. Once the monkey's gaze exited the fixation window (4° square window), the kinematogram was extinguished. Once the monkey's gaze reached the choice target window (4° square window), a 400 ms minimum fixation time was imposed to register the monkey's choice. Correct choices were rewarded with juice. Error choices were not rewarded and penalized with a timeout before the next trial (3 s for monkey F, 0.5–2 s for monkey C).

Two asymmetric reward contexts were alternated in a block design. In Contra-LR blocks, the choice contralateral to the recording/stimulation site was paired with large reward (LR). In Ipsi-LR blocks, the choice ipsilateral to the recording/stimulation site was paired with large reward. The other choice was paired with small reward. At the start of each block, the choice targets were presented with different colors to signal the current reward context to the monkeys, followed by two additional high-coherence trials to allow the monkeys to experience the current reward context. These trials were excluded from analysis. For recording sessions, each block consisted of ~49 trials for monkey C (IQR: 43–55) and 40 trials for monkey F (IQR: 36–55). For microstimulation sessions, each block consisted of ~100 trials (IQR: 98–118) for monkey C and ~61 trials (IQR: 60–61) for monkey F. The large:small reward ratio was ~2:1 (2.02 ± 0.19) for monkey C for both recording and microstimulation sessions. For monkey F, the large:small reward ratio was 3:2 (1.48 ± 0.13) for recording sessions and 5:3 (1.65 ± 0.03) for microstimulation sessions. For monkey C, the average small reward was 0.1 and 0.07 mL and the average large reward was 0.2 and 0.12 mL during recording and microstimulation sessions, respectively. For monkey F, the average small reward was 0.2 and 0.12 mL and the average large reward was 0.29 and 0.25 mL during recording and microstimulation sessions, respectively.

### Data acquisition

Eye position was monitored using a video-based system (ASL) sampled at 240 Hz. Single-unit recordings focused on putative projection neurons (*Ding and Gold, 2010*). We searched for task-relevant neurons while the monkeys performed the equal-reward motion discrimination task with horizontal dots motions and determined the presence of task-related modulation of neural activity by visual and audio inspection of ~10–20 trials. For analyses of neural response properties in recording sessions, only well-isolated single units were included. For analyses of microstimulation effects, sites with either single- or multi-unit task-related modulations were used. Neural signals were amplified, filtered and stored using a MAP acquisition system (Plexon, Inc), along with time-stamped event codes, analog eye position signals and trial parameter values. Single-unit activity was identified by offline spike sorting (Offline Sorter, Plexon, Inc). Multi-unit activity was measured using waveforms that passed an offline amplitude threshold. For the microstimulation experiments, we first identified a caudate site with task-related activity and then interleaved trials with and without microstimulation pseudo-randomly at a 1:1 ratio. Electrical microstimulation was delivered during motion stimulus presentation (negative-leading bipolar current pulses, 300 Hz, 50 µA, 250 µs pulse duration; from motion onset until a saccade was detected). Caudate microstimulation with these parameters did not evoke saccades (*Ding and Gold, 2012b*; *Nakamura and Hikosaka, 2006a*; *Watanabe and Munoz, 2010*).

## Quantification and statistical analysis

### Neural data analysis

For each single/multi-unit dataset, we computed the average firing rates in seven task epochs (*Figure 1A*): three epochs before motion stimulus onset (Epoch 1: 400 ms window beginning at target onset, Epoch 2: a variable window from target onset to motion onset, and Epoch 3: a 400 ms window ending at motion onset), two epochs during motion viewing (Epoch 4: a fixed window from 100 ms after motion onset to 100 ms before median RT, and Epoch 5: a variable window from 100 ms after motion onset to 100 ms before saccade onset), a peri-saccade 300 ms window beginning at 100 ms before saccade onset (Epoch 6), and a post-saccade 400 ms window beginning at saccade onset (Epoch 7; before feedback and reward delivery). For each unit, a multiple linear regression was performed on the firing rates in correct trials, for each task epoch separately.

$$
\begin{aligned}
\mathrm{FR} \ =\ & \beta_0 + \beta_{\mathrm{Choice}} \times \mathrm{I}_{\mathrm{Choice}} + \beta_{\mathrm{RewCont}} \times \mathrm{I}_{\mathrm{RewCont}} + \beta_{\mathrm{RewSize}} \times \mathrm{I}_{\mathrm{RewSize}} \\
& + \ \beta_{\mathrm{Coh-Contra}} \times \mathrm{I}_{\mathrm{Coh-Contra}} + \beta_{\mathrm{Coh-Ipsi}} \times \mathrm{I}_{\mathrm{Coh-Ipsi}} \\
& + \ \beta_{\mathrm{RewCoh-Contra}} \times \mathrm{I}_{\mathrm{Coh-Contra}} \times \mathrm{I}_{\mathrm{RewSize}} + \beta_{\mathrm{RewCoh-Ipsi}} \times \mathrm{I}_{\mathrm{Coh-Ipsi}} \times \mathrm{I}_{\mathrm{RewSize}},
\end{aligned}
\tag{1}
$$

where

$$
\mathrm{I}_{\mathrm{Choice}} = \begin{cases} 1 \text{ for contralateral choice} \\ -1 \text{ for ipsilateral choice} \end{cases},
$$

$$
\mathrm{I}_{\mathrm{RewCont}} = \begin{cases} 1 \text{ for contralateral} - \text{large reward blocks} \\ -1 \text{ for ipsilateral} - \text{large reward blocks} \end{cases},
$$

$$
\mathrm{I}_{\mathrm{RewSize}} = \begin{cases} 1 \ \ \text{if a large reward is expected for the choice} \\ -1 \ \ \text{if a small reward is expected for the choice} \end{cases},
$$

$$
\mathrm{I}_{\mathrm{Coh-Contra}} = \begin{cases} \text{absolute coherence for contralateral choice (centered at mean value)} \\ 0 \text{ for ipsilateral choice} \end{cases},
$$

and

$$
\mathrm{I}_{\mathrm{Coh-Ipsi}} = \begin{cases} 0 \text{ for contralateral choice} \\ \text{absolute coherence for ipsilateral choice (centered at mean value)} \end{cases}.
$$

Significance of non-zero coefficients was assessed using *t*-test (criterion: p=0.05).

## Behavioral analysis

### Measuring microstimulation effects on choice using a logistic function

For each microstimulation session, a logistic function was fitted to the choice data for all trials:

$$P_{\text{contra choice}} = \frac{1}{1 + e^{-\text{Slope} \times (\text{Coh} + \text{Bias})}}, \tag{2}$$

where $\text{Coh}$ is the signed motion coherence,

$$\text{Slope} = \text{slope0} + \text{slope}_{\text{rew}} \times \text{RewCont} + \text{slope}_{\text{estim}} \times \text{Estim} + \text{slope}_{\text{rew} \times \text{estim}} \times \text{RewCont} \times \text{Estim},$$

$$\text{Bias} = \text{bias0} + \text{bias}_{\text{rew}} \times \text{RewCont} + \text{bias}_{\text{estim}} \times \text{Estim} + \text{bias}_{\text{rew} \times \text{estim}} \times \text{RewCont} \times \text{Estim},$$

$$\text{RewCont} = \begin{cases} 1 \text{ for contralateral} - \text{large reward blocks} \\ -1 \text{ for ipsilateral} - \text{large reward blocks} \end{cases},$$

and

$$\text{Estim} = \begin{cases} 1 \text{ for microstimulation trials} \\ 0 \text{ for control trials} \end{cases}.$$

Significance of non-zero coefficients was tested using bootstrap methods by shuffling *Estim* values across trials for each session 200 times (criterion: the experimental value is outside the 95% confidence intervals for shuffled data).

### Measuring microstimulation effects on RT using a linear function

RT was measured as the time from motion onset to saccade onset, the latter identified offline with respect to velocity ($>40°/s$) and acceleration ($>8000°/s^2$). A linear function was fitted to RT data for correct trials:

$$\begin{aligned} \text{RT} = {} & \text{rt0} + \text{Intercept}_{\text{rew}} \times \text{RewCont} + \text{Slope}_{\text{rew}} \times \text{RewCont} \times \text{Coh}_{\text{abs}} \\ & + \text{Intercept}_{\text{estim}} \times \text{Estim} + \text{Slope}_{\text{estim}} \times \text{Estim} \times \text{Coh}_{\text{abs}} \\ & + \text{Intercept}_{\text{rew} \times \text{estim}} \times \text{RewCont} \times \text{Estim} + \text{Slope}_{\text{rew} \times \text{estim}} \times \text{RewCont} \times \text{Estim} \times \text{Coh}_{\text{abs}}, \end{aligned} \tag{3}$$

where $\text{Coh}_{\text{abs}}$ is the absolute value of motion coherence. Significance of non-zero coefficients was tested using *t*-test (criterion: $p = 0.05$).

### Measuring microstimulation effects on both choice and RT using the drift-diffusion model (DDM)

We also fitted the choice and RT data for all trials simultaneously to different variants of the drift-diffusion model (DDM; *Figure 4A*). The basic DDM assumed that the decision variable (DV) is the time integral of evidence (*E*), which was modeled as a Gaussian distributed random variable,

$$E \sim N(k * coherence,\ 1) \text{ and } DV = \int E\, dt$$

The scale parameter *k* controlled the drift rate. At each time point, the DV was compared with two collapsing choice bounds (*Zylberberg et al., 2016*). The time course of the choice bounds was specified as $a/(1 + e^{\beta\_alpha(t - \beta\_d)})$, where $\beta\_alpha$ and $\beta\_d$ controlled the rate and onset of decay, respectively. Note that the effects of collapsing bounds can be equivalently implemented by adding a choice-independent urgency signal (*Drugowitsch et al., 2012*; *Churchland et al., 2008*). If DV crossed the upper bound first, a contralateral choice was made; if DV crossed the lower bound first, an ipsilateral choice was made. RT was modeled as the sum of the time till first bound crossing and saccade-specific non-decision times that accounted for evidence-independent sensory/motor delays (*t_contra* and *t_ipsi*). Two types of biases were used to account for reward asymmetry-induced biases, a bias in drift rate (*me*) and a bias in the relative bound heights (*z*) (*Fan et al., 2018*).

DDM model fitting was performed, separately for each session, using the maximum a posteriori estimate method (python v3.5.1, pymc 2.3.6) and prior distributions suitable for human and monkey

subjects (*Wiecki et al., 2013*). We performed at least five runs for each variant and used the run with the highest likelihood for further analyses.

We used eight variants of the DDM model: in the "Full" model, all parameters were allowed to vary by reward context and microstimulation status; in the "NoEstim" model, all parameters were allowed to vary by reward context, but not microstimulation status (*Figure 5—figure supplement 2A*); in the "NoCollapse" model, *β_alpha* and *β_d* were fixed across microstimulation status; in the "NoA", "NoK", "NoME", and "NoZ" models, *a, k, me,* and *z* were fixed across microstimulation status, respectively; and in the "NoT0" model, saccade-specific non-decision times (*t_contra* and *t_ipsi*) were fixed across microstimulation status (*Figure 5—figure supplement 2B and C*). We used the Akaike information criterion (AIC) for model comparisons, with lower values indicating more parsimonious model variants.

For a given DDM parameter (*Para*), we parsed the different effects as follows (LR: large reward):

$$\Delta \text{Para(base)} = \left(\text{Para}_{\text{contra}-\text{LR, no estim}} + \text{Para}_{\text{ipsi}-\text{LR, no estim}}\right)/2$$

$$\Delta \text{Para(rew)} = \left(\text{Para}_{\text{contra}-\text{LR, no estim}} - \text{Para}_{\text{ipsi}-\text{LR, no estim}}\right)/2$$

$$\Delta \text{Para(estim)} = \left(\text{Para}_{\text{contra}-\text{LR,estim}} + \text{Para}_{\text{ipsi}-\text{LR,estim}} - \text{Para}_{\text{contra}-\text{LR,no estim}} - \text{Para}_{\text{ipsi}-\text{LR, no estim}}\right)/2$$

$$\Delta \text{Para(rew} \times \text{estim)} = \left(\text{Para}_{\text{contra}-\text{LR,estim}} - \text{Para}_{\text{ipsi}-\text{LR,estim}} - \text{Para}_{\text{contra}-\text{LR,no estim}} + \text{Para}_{\text{ipsi}-\text{LR,no estim}}\right)/2$$

## Generating predictions for hypothesized reward context-microstimulation interaction effects on drift rate and bound height

We simulated four types of microstimulation effects (*Figure 4*). For all simulations, we assumed that,

$$\Delta bound(rew) = \gamma \times \Delta drift(rew) + \delta,$$

$$\mu_{drift} = mean(\Delta drift(rew \times estim)), \sigma_{drift} = std(\Delta drift(rew \times estim)),$$

$$\mu_{bound} = mean(\Delta bound(rew \times estim)), \sigma_{bound} = std(\Delta bound(rew \times estim)),$$

where $\gamma$ and $\delta$ are the slope and intercept measured from linear regression of Δdrift (rew) and Δbound (rew) in the experimental data (*Figure 5B*). Δbound (rew) for different sessions (colors in *Figure 4B*) were set as equally spaced values within the range for the experimental data.

For the simulations in *Figure 4C*,

$$\Delta drift(rew \times estim) \sim Normal(0, \sigma_{drift}),$$

$$\Delta bound(rew \times estim) \sim Normal(0, \sigma_{bound}),$$

For the simulation in *Figure 4D*,

$$\Delta drift(rew \times estim) \sim Normal(\mu_{drift}, \sigma_{drift}),$$

$$\Delta bound(rew \times estim) \sim Normal(\mu_{bound}, \sigma_{bound}),$$

For the simulation in *Figure 4E*,

$$\Delta drift(rew \times estim) \sim Normal(\mu_{drift}, \sigma_{drift}),$$

$$\Delta bound(rew \times estim) \sim \gamma \times \Delta drift(rew \times estim) + Normal(0, \sigma_{bound} - \gamma \times \sigma_{drift}),$$

For the simulation in *Figure 4F*,

$$\Delta drift(rew \times estim) \sim k \times \Delta drift(rew) + m + Normal(0, \sigma_{drift}/2),$$

$$\Delta bound(rew \times estim) \sim \gamma \times \Delta drift(rew \times estim) + Normal(0, \ \sigma_{bound}/2),$$

where $k$ and $m$ are slope and intercept, respectively, measured from linear regression of $\Delta drift(rew \times estim)$ and $\Delta drift(rew)$ in the experimental data. The noise terms are reduced to account for variations in $\Delta drift(rew)$.

## Δdrift (rew × estim) and Δbound (rew × estim)

We performed three shuffling-based control analyses (*Figure 6*). For data shown in *Figure 6B–D*, we pooled the fitted DDM values across reward-microstimulation conditions and sessions for each parameter, resampled from these values with replacement for each session and reward-microstimulation condition, simulated sessions with matched numbers of trials for each condition using the resampled parameter values, and re-fitted these simulated data with the Full and NoCollapse DDM models ("Shuffle 1"). For data shown in *Figure 6E*, we performed 200 iterations of shuffling for three types of schemes. For "Shuffle 2", we shuffled the fitted $me$ and $z$ values for trials with and without microstimulation across sessions. For "Shuffle 3", we kept the $me$ and $z$ values for trials without microstimulation and shuffled $me$ and $z$ values independently for trials with microstimulation. For "Shuffle 4", we kept the $me$ and $z$ values for trials without microstimulation and shuffled the session identity for paired $me$ and $z$ values for trials with microstimulation. The fitted values from the simulated/shuffled data were parsed in the same way as those from the experimental data. We considered the experimental result to be significantly different from shuffled results if the former was outside the 95% confidence intervals of the latter.

## Relating microstimulation effects to neural selectivity and the monkeys' voluntary strategy

For the microstimulation experiments, we recorded single- or multi-unit activity before microstimulation and performed the multiple linear regression in *Equation 1* for each unit separately. The regression coefficients were normalized by $\beta_0$ in the regression, which measured the average firing rates. These $\beta$ values indexed the neural selectivity for different task-related regressors. If more than one unit was recorded at a site, for each regressor, we used the $\beta$ value associated with the lowest $p$ value (i.e. most reliable modulation).

We derived two principal components (PCs) from Δdrift (rew, no estim) and Δbound (rew, no estim) values (*Figure 7A and B*, left). The projections of Δdrift (rew, no estim) and Δbound (rew, no estim) values onto the two PCs indexed the effects of reward context on the monkeys' voluntary strategy in each session, relative to their average tendencies across sessions.

We projected Δdrift (rew × estim) and Δbound (rew × estim) values onto the same two PCs (*Figure 7A and B*, right). These projections indexed the microstimulation effects in each session, relative to the monkeys' average tendencies in their reward context-dependent voluntary strategy across sessions. We then performed the following multiple linear regression:

$$
\begin{aligned}
\text{Projection of (rew} \times \text{estim) values} = &\ \alpha_0 + \beta_{\text{Choice}} \times \alpha_{\text{Choice}} + \beta_{\text{RewCont}} \times \alpha_{\text{RewCont}} + \beta_{\text{RewSize}} \times \alpha_{\text{RewSize}} \\
&+ \beta_{\text{Coh-Contra}} \times \alpha_{\text{Coh-Contra}} + \beta_{\text{Coh-Ipsi}} \times \alpha_{\text{Coh-Ipsi}} \\
&+ \beta_{\text{CohRew-Contra}} \times \alpha_{\text{CohRew-Contra}} + \beta_{\text{CohRew-Ipsi}} \times \alpha_{\text{CohRew-Ipsi}} \\
&+ \text{Projection of (rew, no estim) values} \times \alpha_{\text{projection}}
\end{aligned}
\tag{4}
$$

The $\alpha$ values (except for $\alpha_0$) measured the dependence of microstimulation effects on neural selectivity and the monkeys' daily variations in their voluntary strategy. Significance of $\alpha$ values was assessed by comparing the experimental value with their corresponding null distributions. The null distributions were estimated using 1000 iterations of regression based on shuffled independent variables. The number of iterations with values exceeding the experimental values was used to estimate $p$ values for regression coefficients (*Figure 7C*) and explained variance (*Figure 7D*). A criterion of $p<0.0125$ was used to correct for multiple regressions (0.05/4 regressions).

## Acknowledgements

We thank Jean Zweigle for animal care and Drs. Kae Nakamura and Ethan Bromberg-Martin for help-ful comments. This work was supported by NIH National Eye Institute (R01-EY022411; LD and JIG), University of Pennsylvania (University Research Foundation Pilot Award; LD), and Hearst Foundations Graduate student fellowship (YF).

## Additional information

### Competing interests

Joshua I Gold: Senior editor, *eLife*. The other authors declare that no competing interests exist.

### Funding

| Funder | Grant reference number | Author |
|---|---|---|
| National Eye Institute | R01-EY022411 | Joshua I Gold<br>Long Ding |
| Hearst Foundations | Graduate Student Fellowship | Yunshu Fan |
| University of Pennsylvania | University Research Foundation Pilot Award | Long Ding |

The funders had no role in study design, data collection and interpretation, or the decision to submit the work for publication.

### Author contributions

Takahiro Doi, Yunshu Fan, Data curation, Formal analysis, Investigation, Visualization, Methodology, Writing - review and editing; Joshua I Gold, Funding acquisition, Visualization, Writing - review and editing; Long Ding, Conceptualization, Resources, Data curation, Formal analysis, Supervision, Fund-ing acquisition, Validation, Investigation, Visualization, Methodology, Writing - original draft, Project administration, Writing - review and editing

### Author ORCIDs

Takahiro Doi https://orcid.org/0000-0002-9650-972X
Yunshu Fan https://orcid.org/0000-0003-2597-5173
Joshua I Gold https://orcid.org/0000-0002-6018-0483
Long Ding https://orcid.org/0000-0002-1716-3848

### Ethics

Animal experimentation: All training and experimental procedures were in accordance with the National Institutes of Health Guide for the Care and Use of Laboratory Animals and were approved by the University of Pennsylvania Institutional Animal Care and Use Committee (protocol #804726).

### Decision letter and Author response

Decision letter https://doi.org/10.7554/eLife.56694.sa1
Author response https://doi.org/10.7554/eLife.56694.sa2

## Additional files

### Supplementary files

• Supplementary file 1. a, Median and p values for microstimulation-induced effects for all 55 sites, as measured by logistic fits to the choice data and linear fits to the RT data.P values were from Wil-coxon signed rank test. Bold: p<0.05. b, Median and p values for microstimulation-induced effects in

39 effective sites, as measured by best DDM fits to the choice and RT data. P values were from Wilcoxon signed rank test. Bold: p<0.05.

- Transparent reporting form

### Data availability

Source data are uploaded for main data figures/tables (i.e. Figure 2, Figure 3 and figure supplement 1, Figure 5 and its figure supplements 2 and 4, Figure 6A and E, Figure 6 -figure supplement 3, Figure 7, and Supplementary file 1).

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
