## [Decision Letter]

**Acceptance summary:**

The authors showed that heterogeneous activity of neurons in the caudate nucleus reflected all major aspects during perceptual decision with asymmetric rewards. They also found that the effect of electrical stimulation in the caudate nucleus was quantitatively related not only to the behavioral strategy of the animal but also to the functional properties of the neurons recorded in the stimulation site. These results establish the role of the caudate nucleus in mediating the reward bias during perceptual decision making.

**Decision letter after peer review:**

Thank you for submitting your article "The caudate nucleus contributes causally to decisions that balance reward and uncertain visual information" for consideration by *eLife*. Your article has been reviewed by three peer reviewers, including Daeyeol Lee as the Reviewing Editor and Reviewer #1, and the evaluation has been overseen by Michael Frank as the Senior Editor. The following individual involved in review of your submission has agreed to reveal their identity: Chandramouli Chandrasekaran (Reviewer #3).

The reviewers have discussed the reviews with one another and the Reviewing Editor has drafted this decision to help you prepare a revised submission.

As the editors have judged that your manuscript is of interest, but as described below that some additional analyses might be required before it is published, we would like to draw your attention to changes in our revision policy that we have made in response to COVID-19 (https://elifesciences.org/articles/57162). First, because many researchers have temporarily lost access to the labs, we will give authors as much time as they need to submit revised manuscripts. We are also offering, if you choose, to post the manuscript to bioRxiv (if it is not already there) along with this decision letter and a formal designation that the manuscript is 'in revision at *eLife*'. Please let us know if you would like to pursue this option. (If your work is more suitable for medRxiv, you will need to post the preprint yourself, as the mechanisms for us to do so are still in development.)

Summary:

This study examined the role of the caudate nucleus in combining uncertain sensory information with reward bias during decision making by analyzing the effects of electrical stimulation and neural activity within the frame of a drift-diffusion model. In their previous study, the authors have already characterized the biasing effect of unequal reward on perceptual decision making using two parameters corresponding to the change in the drift rate (*me*) and bound (*z*). There are at least two important findings in this study. First, they showed that heterogeneous activity of neurons in the caudate nucleus reflected all major aspects of the task, including coherence (i.e., signal-to-noise ratio), choice, and reward bias, and their interactions. Second, they found that the effect of electrical stimulation was quantitatively related not only to the behavioral strategy of the animal but also to the functional properties of the neurons recorded in the stimulation site. While these results seem to firmly establish the role of the caudate nucleus in mediating the reward bias during perceptual decision making, there is a room for improvement in the analyses and presentation. The authors should improve their exposition in the modeling section and to simplify their results where possible to make it easier on the reader. A large number of parameters introduced in the paper makes it difficult to keep track of everything.

Essential revisions:

1) The section that describes the relationship between the effect of microstimulation and the animal's strategy (subsection “Microstimulation caused coordinated adjustments to reward-dependent decision biases that mimicked the monkeys’ voluntary strategy”) was very hard to follow and is not entirely convincing. This section relies heavily on the findings from the previous study by the authors that the negative correlation between the changes in the drift and bound reflected the animal's heuristic strategy for reward maximization. The results in that study also suggest that such adjustment occurred slowly across many sessions. Was it the case then that the effect of stimulation changed similarly during the same period? If so, the observed relationship between the animal's strategy and stimulation effect might be spurious and caused by a slow cross-session drift (any slow changes in the animal's strategy not captured by the model).

2) Microstimulation result in Figure 3 is complex to parse. There are 12 distributions with 6 panels of scatter plots. Might it make sense to perhaps obtain a figure that demonstrates the effect size for rew x estim. Appropriately normalized regressions might allow such a plot. That way, one could understand the magnitude of the effect.

3) The modeling in Figures 4 and 5 are nice. The coordinated additive and scaling hypothesis assume that the monkeys change bound height and drift rate across sessions. Microstimulation has a coordinated effect on Δbound and Δdrift. The data in Figure 5 is largely consistent with the Figure 4E and F. However, the results are not entirely convincing, in part because they entirely depend on the contrasts in the model parameters for reward context dependent and independent components. However, if the underlying parameters (e.g., Para_contra-LR_,_estim_, etc.) are correlated, the results might need a different interpretation. It might be possible to address this, using PCA. For example, if the authors run PCA on the parameters used to quantify the effect of reward without stimulation (e.g., Para_contro-LR,no estim_) and PAC on parameters used to quantify ΔPara(estim) and ΔPara(rew x estim), do they find significant correlation for more than one PC?

There are also some discrepancies between the effect of microstimulation presented in Figure 4E, F. First microstimulation leads to a shift diagonally upwards from the line in Figure 5D which is inconsistent with Figure 4C and F. Whereas the coordinated additive and coordinated scaling push it downward. This needs to be better explained to ameliorate concerns for the reader. For example, placing the scatter plots from Figure 4C-F middle and right columns and placing them in Figure 5 might be helpful. The colormap should also be the same between Figures 4 and 5 so that one can compare both of these reliably. They could sacrifice some of the logistic vs. DDM plots to the supplements. There are several additional concerns.

1) Model predicts more lower Δdrift and lower Δbound for the coordinated scaling case, but the data don't seem to show that. Is this important?

2) What happens when you change the size of the noise for Figure 4D, does it show a less steeper change in Figure 4D right panel?

3) Quantitative evidence brought to bear to favor the coordinated scaling over the independent model in Figure 4D right panel would make it more compelling. Currently, one can look at Figure 4F and Figure 5F and see they are similar.

4) The fact that correlation coefficient (slope) between changes in drift and bound (Figure 6E) tends to be negative for Shuffles 3 and 4 is a concern, because this raises the possibility that the observed relationship between the two measures might be also an artifact. Can this be addressed using a simpler approach of shuffling the parameters fit for different stimulation and reward context (e.g., Para_contra-LR,estim_), rather than simulating individual trials and re-estimating the model parameters?

5) There are some problems in how neural activity was analyzed using a regression model. For example, the activity during a variable window might be confounded with reaction time (e.g., Epoch 5; subsection “Caudate neurons encode both visual and reward information”, Figure 2—figure supplement 2), which could lead to mis-identification of signals related to coherence. Also, the statement that a majority of the neurons showed at least one of many effects is not particularly meaningful, since the proportion of such neurons is expected to increase with the number of tests when they were not corrected for multiple comparisons. Similarly, the effects of microstimulation on multiple types of behavioral data (subsection “Caudate microstimulation evoked reward context-dependent effects on behavior”) should be reported more carefully.

6) Some of the results reported in this study might require more careful interpretation. For example, the fact that microstimulation effected varied with the reward context by itself might not provide strong evidence that the caudate nucleus is causally involved in balancing visual evidence and reward bias, because such interaction might occur when the magnitude of microstimulation varies with the reaction time (which is in turn influenced by reward context).

---

## [Author Response]

Essential revisions:1) The section that describes the relationship between the effect of microstimulation and the animal's strategy (subsection “Microstimulation caused coordinated adjustments to reward-dependent decision biases that mimicked the monkeys’ voluntary strategy”) was very hard to follow and is not entirely convincing. This section relies heavily on the findings from the previous study by the authors that the negative correlation between the changes in the drift and bound reflected the animal's heuristic strategy for reward maximization. The results in that study also suggest that such adjustment occurred slowly across many sessions. Was it the case then that the effect of stimulation changed similarly during the same period? If so, the observed relationship between the animal's strategy and stimulation effect might be spurious and caused by a slow cross-session drift (any slow changes in the animal's strategy not captured by the model).

The reviewers are correct that our previous study showed that the negative correlation between the reward context-dependent adjustments in drift rates and bounds reflected the animal’s heuristic strategy for achieving good-enough reward outcomes. However, as we showed in that paper, the monkeys’ adjustments reflected the reward functions for the two types of blocks in a given session, arguing for fast adaptation that occurred within a block of trials, instead of slow adaptation across many sessions. As shown in the Author response image 1, the monkeys’ adjustments on no-estim trials varied from session to session, but with no clear trend across sessions. We thus do not believe that the alternative explanation raised by the reviewers could explain our estim data. We have clarified this point in the main text, including the following: "The monkeys tended to use positive ∆drift (rew) values that changed in magnitude according to session-by-session variations in the reward function (see Figure 6 in (Fan et al., 2018)), implying fast adaptation within a session."

**Author response image 1. sa2fig1:** Monkeys did not show trends in their drift/bound biases across sessions. Data were from trials without microstimulation in microstimulation sessions.

2) Microstimulation result in Figure 3 is complex to parse. There are 12 distributions with 6 panels of scatter plots. Might it make sense to perhaps obtain a figure that demonstrates the effect size for rew x estim. Appropriately normalized regressions might allow such a plot. That way, one could understand the magnitude of the effect.

We apologize for the dense figure. We now present the summary histograms and cartoon illustrations of the effects of these parameters in the main figure and present scatterplots of individual sessions in Figure 3—figure supplement 1. We hope these revisions make the admittedly complex results more accessible to the readers.

3) The modeling in Figures 4 and 5 are nice. The coordinated additive and scaling hypothesis assume that the monkeys change bound height and drift rate across sessions. Microstimulation has a coordinated effect on Δbound and Δdrift. The data in Figure 5 is largely consistent with the Figure 4E and F. However, the results are not entirely convincing, in part because they entirely depend on the contrasts in the model parameters for reward context dependent and independent components. However, if the underlying parameters (e.g., Para_contra-LR,estim_, etc.) are correlated, the results might need a different interpretation. It might be possible to address this, using PCA. For example, if the authors run PCA on the parameters used to quantify the effect of reward without stimulation (e.g., Para_contro-LR,no estim_) and PAC on parameters used to quantify ΔPara(estim) and ΔPara(rew x estim), do they find significant correlation for more than one PC?

The reviewers raised a valid concern about whether the coordination we observed in Δdrift (rew × estim) and Δbound (rew × estim) represented the primary feature in the experimental data, or secondary to other correlation patterns. We thank the reviewers for suggesting the PCA-based analyses to address this concern. We performed PCA on the eight raw fitted parameters ([*me* or *z*] × [contra-LR or ipsi-LR] × [with estim or no estim]). We then examined the relationship between the first PC from this analysis

(“PC1 (8 para)”) and the coordination of Δdrift (rew × estim) and Δbound (rew × estim), the latter being quantified as the first PC (“PC1 (coord)”) in the two-parameter space. We projected the fitted parameters from the experimental data onto the PC1 (8para) and PC1 (coord), respectively, and found a strong correlation between the projection values (rho = 0.92, p = 1.3e-16). These results strongly suggest that the coordinated effects were the primary feature of the experimental data; i.e., the dominant effects of caudate microstimulation. We now include these results in Figure 6—figure supplement 3 and text:

“Fourth, given the inter-session variability in microstimulation effects, it is possible that the negatively correlated ∆drift (rew × estim) and ∆bound (rew × estim) reflected only a minor consequence of caudate microstimulation. […] The projections showed a strong correlation (Figure 6—figure supplement 3; rho=0.92, *p*=1.3e-16), indicating that the coordinated ∆drift (rew × estim) and ∆bound (rew × estim) values reflected the dominant feature of the experimental data; i.e., the dominant effect of caudate microstimulation on reward-related biases.”

There are also some discrepancies between the effect of microstimulation presented in Figure 4E, F. First microstimulation leads to a shift diagonally upwards from the line in Figure 5D which is inconsistent with Figure 4C and F. Whereas the coordinated additive and coordinated scaling push it downward. This needs to be better explained to ameliorate concerns for the reader. For example, placing the scatter plots from Figure 4C-F middle and right columns and placing them in Figure 5 might be helpful. The colormap should also be the same between Figures 4 and 5 so that one can compare both of these reliably. They could sacrifice some of the logistic vs. DDM plots to the supplements. There are several additional concerns.1) Model predicts more lower Δdrift and lower Δbound for the coordinated scaling case, but the data don't seem to show that. Is this important?2) What happens when you change the size of the noise for Figure 4D, does it show a less steeper change in Figure 4D right panel?3) Quantitative evidence brought to bear to favor the coordinated scaling over the independent model in Figure 4D right panel would make it more compelling. Currently, one can look at Figure 4F and Figure 5F and see they are similar.

We again apologize for the confusion. Figure 4 aimed to illustrate the idea that coordinated changes would predict correlated rew x estim terms, whereas independent changes would not. The parameters used for these illustrations were arbitrarily picked just for illustration purposes and not fitted to the experimental data. This is why the simulated data points in Figure 4F were still somewhat mismatched from the experimental data in Figure 5.

We understand the reviewers’ concern and have generated new illustrations with parameters that were based on the experimental data, with comparable magnitudes of both stimulation effects and noise. As expected, the newly simulated data in Figure 4F much more closely match the experimental data (e.g., the fitted line in Figure 4F is within the 95% confidence intervals of the fitted line in Figure 5D), without the concerning features that were noted in the concerns (1) and (2).

For the concern (3), we believe that we have already provided this quantitative evidence: the “Shuffle 2” test in Figure 6E, which shows that, even with the mean and variance values matched exactly to the experimental data, the model in Figure 4D cannot capture the correlated pattern we observed in the experimental data. We now indicate clearly in the text that this shuffle rules out the model in Figure 4D. In contrast, the correlation is built-in for the model in Figure 4F.

They could sacrifice some of the logistic vs. DDM plots to the supplements.

We followed the reviewers’ suggestion and moved these plots to the supplements. To further simply the figures, we have removed the panels related to “rew effects with estim” in Figures 4 and 5, which are not essential, nor informative, for understanding the stimulation effects.

4) The fact that correlation coefficient (slope) between changes in drift and bound (Figure 6E) tends to be negative for Shuffles 3 and 4 is a concern, because this raises the possibility that the observed relationship between the two measures might be also an artifact. Can this be addressed using a simpler approach of shuffling the parameters fit for different stimulation and reward context (e.g., Para_contra-LR,estim_), rather than simulating individual trials and re-estimating the model parameters?

We apologize for the confusion. Shuffles 3 and 4 represent partial shuffles that maintained different features of the experimental data and thus were expected to maintain some relationship between the two parameters. Even so, the resulting correlations were significantly weaker than in the experimental data.

The full shuffling of individual parameters, as the reviewers suggested, resulted in a distribution that is centered at zero and similar to the “Shuffle 2” data in Figure 6E (when we fully shuffled Δdrift (rew) and Δbound (rew) for both stimulation conditions). To highlight the importance of shuffle 2 for our interpretation, we changed the color codes for the three types of shuffles in Figure 6E. We also revised the text to make these points more clearly:

“Second, the coordinated effects of microstimulation on ∆drift and ∆bound might have been a trivial consequence of reward-dependent biases, independent of the reward context-dependent coordination of the two quantities. […] That is, even with the mean and variance values matched exactly to the experimental data, the model in Figure 4D cannot capture the correlated pattern we observed in the experimental data; and 3) partial shuffling that disrupted only possible relationships with session-specific properties (e.g., microstimulation sites or voluntary performance), by shuffling the paired ∆drift and ∆bound across sessions, also significantly weakened the correlation between “rew × estim” effects (Figure 6E, “Shuffle 4”).”

5) There are some problems in how neural activity was analyzed using a regression model. For example, the activity during a variable window might be confounded with reaction time (e.g., Epoch 5; subsection “Caudate neurons encode both visual and reward information”, Figure 2—figure supplement 2), which could lead to mis-identification of signals related to coherence.

We agree with the reviewers about this concern. This was why, in addition to the variable window (epoch 5), we also analyzed activity in a fixed-duration window (epoch 4 in Figure 2C-F). The results from the two epochs are qualitatively similar, with epoch 5 showing a larger fraction of neurons. These results suggest that the coherence modulation was not simply a result of a dependence on reaction time. We now comment on this observation: “The selectivity for reward size and motion strength was more prevalent in Epoch #5 (variable duration covering the whole motion viewing period) than in Epoch #4 (fixed duration covering only the early motion viewing period), consistent with a developing latent decision variable that accumulates evidence over time, increasing sensitivity of the regression analysis with longer analysis windows, and/or additional sensitivity to RT closer to the time of saccade.”

We show the heatmap of significant coefficients for Epoch 5 in Figure 2—figure supplement 2 to illustrate the diversity of neural modulation. To make this point explicit, we revised the text: “These neurons also showed heterogenous modulation patterns, as illustrated for the 44 neurons with joint modulation during motion viewing, with heterogeneous modulation patterns (Figure 2—figure supplement 2).”

We agree with the reviewers that more comprehensive description of the neural data, including a direct examination of RT selectivity, is valuable. To that end, we are preparing a separate manuscript with more in-depth analysis of the caudate data, which will include time-dependent regression results, using normalized RT as a regressor, for activity aligned to motion and saccade onset, respectively. However, we agree with the reviewers that the present manuscript needs to be simplified, not expanded, and thus prefer to leave these additional analyses to the other manuscript.

Also, the statement that a majority of the neurons showed at least one of many effects is not particularly meaningful, since the proportion of such neurons is expected to increase with the number of tests when they were not corrected for multiple comparisons.

We assume that this statement referred to “Overall, a majority of neurons (101/142) showed at least one of these forms of joint modulation in at least one epoch (Figure 2F)”. We now support this statement with a statistical test based on the comparison of the observed fraction and a chance level that considers the multiple comparisons. For example, the chance levels are 0.0975 for observing at least one modulation by coh x reward out of two terms for contra and ipsi, respectively, and 0.0095 for observing modulations by both coh and reward (“Coh + Reward”). The chancel level for the fraction in Figure 2F is thus 1 – (1-0.0975) * (1-0.0095) = 0.1061, for a single epoch. The overall chance level for observing at least one epoch with joint modulation, out of 7 epochs, is thus 0.5438. The number count we reported (101 out of 142) was significantly above this chance level (chi-square test, p = 0.0035). We revised the text as follows: “Overall, 101 out of 142 neurons showed at least one of these forms of joint modulation in at least one epoch. This fraction was significantly above chance level, even considering the multiple tests done in 7 epochs (Figure 2F; Chi-squared test, p=0.0035).” We also added to the figure legend that the chance level was “adjusted based on the number of tests used and a 5% chance level for a single test”.

Similarly, the effects of microstimulation on multiple types of behavioral data (subsection “Caudate microstimulation evoked reward context-dependent effects on behavior”) should be reported more carefully.

Following similar logic, we revised the text: “Overall, microstimulation induced at least one statistically reliable effect on choice or RT at 48 of 55 caudate sites (colored dots in Figure 3—figure supplement 1B; significantly above the chance level that was adjusted for multiple comparisons, Chi-squared test, p<1e-5).”

6) Some of the results reported in this study might require more careful interpretation. For example, the fact that microstimulation effected varied with the reward context by itself might not provide strong evidence that the caudate nucleus is causally involved in balancing visual evidence and reward bias, because such interaction might occur when the magnitude of microstimulation varies with the reaction time (which is in turn influenced by reward context).

We agree with the reviewers that, because we used a RT task with contextual modulation of RT and perturbed the caudate nucleus during the whole motion-viewing period, it is possible that the contextual difference in perturbation effects might be due to contextual differences in RT. We examined this alternative explanation and did not find much support in our data. We added Figure 3—figure supplement 2 and stated in the text:

“Because we delivered microstimulation throughout motion viewing for an RT task, the difference in average RT between reward contexts may contribute to “rew × estim”-type microstimulation effects. […] Thus, the dependence of the microstimulation effects on reward context more likely reflected a causal involvement of the caudate nucleus in balancing visual evidence and reward asymmetry information.”